# CamoTSS: analysis of alternative transcription start sites for cellular phenotypes and regulatory patterns from 5' scRNA-seq data

Ruiyan Hou[1], Chung-Chau Hon [2,3] & Yuanhua Huang [1,4,5]

Five-prime single-cell RNA-seq (scRNA-seq) has been widely employed to profile cellular transcriptomes, however, its power of analysing transcription start sites (TSS) has not been fully utilised. Here, we present a computational method suite, CamoTSS, to precisely identify TSS and quantify its expression by leveraging the cDNA on read 1, which enables effective detection of alternative TSS usage. With various experimental data sets, we have demonstrated that CamoTSS can accurately identify TSS and the detected alternative TSS usages showed strong specificity in different biological processes, including cell types across human organs, the development of human thymus, and cancer conditions. As evidenced in nasopharyngeal cancer, alternative TSS usage can also reveal regulatory patterns including systematic TSS dysregulations.

Alternative usage of different gene architectures enables to differential expression of various mRNA isoforms, including alternative transcription start/end sites and alternative splicing (AS) events, such as exon skipping, intron retention, alternative 5' and 3' splice sites[1–3]. The advances of single-cell transcriptomic technologies have provided a powerful tool to detect cellular heterogeneity in gene-level expression by using the sum of all transcripts originating from the same gene[4]. Several studies have developed computational and statistical approaches to detect and quantify alternative splicing at single-cell resolution[5]. Most of these studies focus on the exon-skipping event, commonly by full-length based platforms like Smart-seq2[6–8] and also possibly by UMI-based methods like 10X Genomics Chromium[9,10]. Thanks to the higher throughput, 3' tag-based scRNA-seq in 10x Chromium platform has been broadly adopted to explore gene-level expression. Several groups leveraged the technique characteristic of polyA-biased scRNA-seq and developed computational pipelines to detect alternative 3' end usage, even with potential applicability to detect alternative 5' start site usages[11,12]. In addition, another recent

method, *scraps*, took the advantage of using read 1 (>100 bp) to precisely identify polyadenylation sites at a near-nucleotide resolution in scRNA-seq data by 10X Genomics and other TVN-primed libraries[13].

Alternative transcript start site (TSS) usage is another major mechanism to increase transcriptome diversity and its regulation. Cap analysis gene expression (CAGE) has been widely used to capture the 5'-end of transcripts and identify TSS at a single-nucleotide resolution from bulk samples[14], which has been used as the major tool to annotate TSS across mammal genomes in the FANTOM project[15] and to reveal narrow shifts of TSS within a single promoter during zebrafish early embryonic development[16]. The analysis of TSS and its alternative usage has been further fueled by the extensive use of RNA-Seq in multiple international consortium projects, including tissue-specific TSS by the Genotype-Tissue Expression (GTEx) data[1], the cell type specific novel TSS by the RAMPAGE project[17] and cancer type specific promoter regulations from a pan-cancer study[18]. More individual studies also evidenced the importance of TSS regulation for different biological functions, e.g., tumor immune interaction in gastric cancer[19], prognosis

[1]School of Biomedical Sciences, University of Hong Kong, Hong Kong, SAR, China. [2]RIKEN Center for Integrative Medical Sciences, Yokohama City, Kanagawa 230-0045, Japan. [3]Graduate School of Integrated Sciences for Life, Hiroshima University, Higashi-Hiroshima, Japan. [4]Department of Statistics and Actuarial Science, University of Hong Kong, Hong Kong, AR, China. [5]Center for Translational Stem Cell Biology, Hong Kong Science and Technology Park, Hong Kong, SAR, China. e-mail: yuanhua@hku.hk

in multiple myeloma[20], and synchronized cell-fate transitions in the yeast gametogenesis program[21].

Recently, attention has also been paid to TSS analysis at a single-cell level, for example, a single-cell version of CAGE, *C1 CAGE*, was introduced to identify TSS and enhancer activity with the original sample multiplexing strategy in the C1TM microfluidic system[22]. This was further extended to capture both 5' and 3' by the single-cell RNA Cap And Tail sequencing (scRCAT-seq) method, where UMI became applicable to further reduce the cost[23]. Besides these specialized methods, conventional platforms, e.g., 10x Genomics have commercial kits for constructing a 5' gene expression library (often with V(D)J dual readouts), where the fragmentation does not happen at sequences close to template switch oligo (TSO), suggesting that this part of sequences is an ideal material to detect transcription start site at a (near-) nucleotide resolution (Fig. 1A). Many sequencing centers keep its default setting of equal length paired-end (e.g., 150 bp), hence capturing the cDNA in read 1. Indeed, some public 5' 10x Genomics datasets on the GEO repository have such information, bringing an open rich resource to re-explore the TSS usage in various biological contexts at the single-cell level. A software suite called SCAFE has already used this type of sc-end5-seq data to de novo detect TSS at a single-cell resolution, but it mainly paid attention to the cis-regulatory elements (CRE, the proxy of the TSS) rather than the alternative transcription start sites[24]. Therefore, there is an urgent demand for tailored methods to analyse such 5' scRNA-seq both efficiently and accurately, especially on alternative TSS usage.

Here, to identify and quantify potential TSSs and evaluate their differential usages from 5' tag-based scRNA-seq, we fully utilize those "rubbish" sequences in read 1 mentioned above (otherwise trimmed before analysis) and developed CamoTSS (<u>Ca</u>p and <u>Mo</u>tif-based <u>TSS</u> modeling from 5' scRNA-seq data), a computational method suite that calls TSSs by combining clustering of reads distribution and classification with predictive features, followed by window sliding technique to denoise the detection of single-nucleotide-resolution TSS. CamoTSS further focused on the analysis of alternative TSS among different cell populations, tissues, development stages and disease contexts by leveraging our upgraded BRIE2[25] as a backend engine. The effective application of our method was demonstrated by using public 5' scRNA-seq data containing pair-ends (i.e. read 1 covering information of cDNA) including adult human cell atlas of 15 major organs, primary nasopharyngeal carcinoma and hyperplastic lymphoid tissue, gastric cancer paratumor and adjacent paratumor tissues and human thymic cell across development and postnatal life, where alternative usages of TSSs were found with strong specificity on cellular states and showed potential to assess their systematic dysregulation.

Of note, CamoTSS is capable of analysing TSS both at a region (around 100 bp) and a single-nucleotide resolution. For the former, we interchangeably use TSS region, TSS cluster or simply TSS, otherwise, we will specifically call the latter CTSS (CAGE tag-defined transcription start site, a concept borrowed from CAGE[16]). We primarily focus on TSS region/cluster analysis and only introduce the CTSS in the analysis of the thymus development data, considering its biological relevance.

## Results

### Overview of CamoTSS pipeline

We developed a stepwise computational method CamoTSS to detect alternative transcription start site clusters and quantify their differential usage utilizing 5' tag-based scRNA-seq data alone (Fig. 1B). In brief, CamoTSS has three main steps after fetching TSS reads for a certain gene from an aligned bam file: (1) clustering of TSS reads with hierarchical clustering (minimum linkage distance: 100 bp by default), (2) filtering TSS clusters by technical thresholds (minimum UMIs: 50; minimum inter-cluster distance: 300 bp) and an embedded classifier to prevent TSS clusters from artefacts by using predictive features (see

next paragraph) and (3) annotating these de-novo TSSs (by its summit position) to known annotations (e.g., GENCODE) optimized by a Hungarian algorithm while if a detected TSS cluster does not cover the optimal known TSS position, it remains called as new-TC or novel-TSS (Fig. 1B; Methods).

Due to strand invasion[26] and sequence biases[27], the classification is a critical step to rule out false positives caused by technical artifacts (Fig. 1C). By using ATAC-seq data from a matched sample as ground truth, we labeled the intersecting TSSs captured from 5' scRNA-seq data as true TSS if it overlaps with a high-confidence ATAC peak or false TSS if it overlaps with a low-confidence ATAC peak or without any ATAC peak. Based on these samples, four reads-based features (unencoded G percentage[24], summit UMI count and cluster standard deviation, UMI counts) were extracted and then feed to logistic regression to do classification. In addition, we also introduced a convolutional neural network model (architecture detail in Methods) to examine how well the pure genomic sequence (+/- 100 bp) predicts the TSS. Here, for deployment and user usage, the four reads-based features were kept as the default setting with a pre-trained logistic regression by using datasets from pluripotent stem cells (iPSC) and human dermal fibroblasts (DMFB) lines. This setting achieves a balance between reasonable accuracy, high generalizability and remarkable simplicity (see next section). On the other hand, we keep the sequence-based model as an optional component not only to enhance prediction performance but can also imply regulatory strength (see the NPC section).

### Performance of CamoTSS in detecting TSS

To assess the performance of our proposed classifiers in filtering false positive artefacts, we leveraged the positive and negative TSSs annotated by matched ATAC data (see above and Methods) and performed tenfold cross-validations. In the combined data of iPSC and DMFB lines, all models achieve high performance in both specificity and sensitivity, with the area under the receiver operating characteristic curve (AUROC or AUC) of 0.976, 0.988, 0.987 and 0.984 for cluster model, sequence model, combined features models with separately-training and jointly-training, respectively (Fig. 2A, Supplementary Fig. S1A; all using same cross-validation split). Similar high accuracy was also observed when running the same assessment on iPSC and DMFB lines separately (Supplementary Fig. S1B, C). To prove the generalizability of both the cluster and sequence models, we further performed the out-of-distribution prediction. Specifically, for the cluster model, we found it generalized well by training on iPSC and predicting DMFB (AUC = 0.985; Fig. 2B) or from the combination of iPSC and DMFB to PBMC (AUC = 0.965; Fig. 2B). For the sequence model, we evidenced its robustness by knocking out the binding site sequences of a certain TF (CTCF or E2F6) in the training set and only predicting this TF's binding sequence as a test set (AUC = 0.908 for CTCF and 0.948 for E2F6; Fig. 2C). We speculate this may be due to the intrinsic similarity of binding sites among different TFs, especially those in a certain family. When checking the importance of features by combining all features using the separately-trained CNN model, we found that the DNA sequence feature set (i.e. 32 weight features from the CNN model; AUC = 0.976) is the most predictive one, followed by a single feature, the unencoded G percentage (AUC = 0.974, Fig. 2D). The coefficients of logistic regression display a consistent importance rank (Supplementary Fig. S2).

Furthermore, the quality of detected TSS clusters can be assessed by the consistency with epigenetic signals, including promoter-specific histone modifications (H3K4me3 and H3K27ac), gene-body-specific markers (H3K36me3)[28] and transcription initiation signal (RNA POL2 that collects general transcription factors to form the pre-initiation complex)[29]. In the PBMC dataset, the RNA POL2, H3K4me3 and H3K37ac signals were all highly enriched in the intervals around CamoTSS-identified TSS clusters (no matter located in an exon or

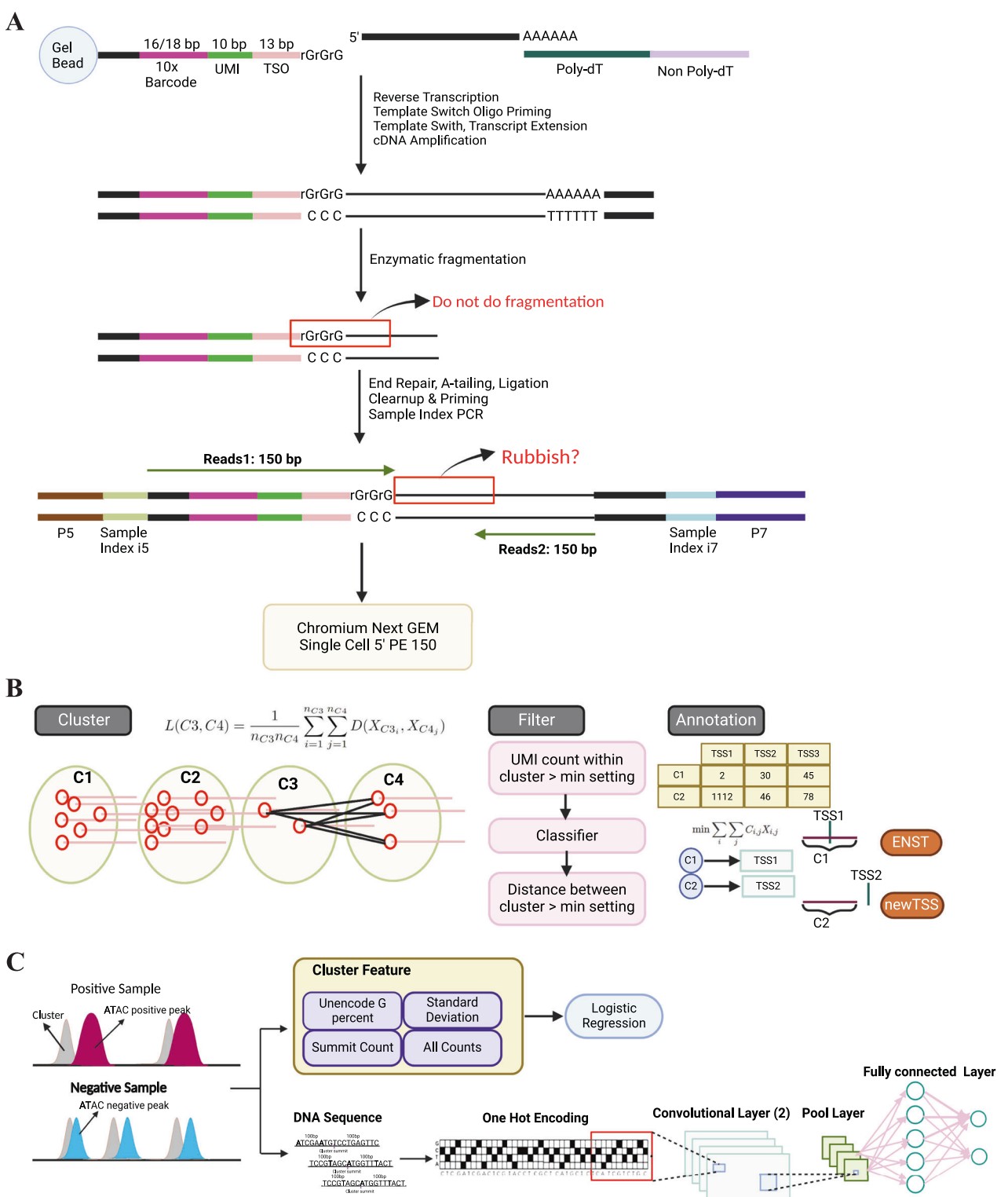

**Fig. 1 | Developing CamoTSS to identify transcription start site (TSS) from 5' tag-based scRNA-seq data. A** A flow chart of the 5' scRNA-seq gene expression library construction (10x Genomics). **B** A schematic of CamoTSS which includes clustering, filtering and annotation. "C" denotes cluster. The lines within the cluster circles represent the aligned reads and their start positions are denoted by red circles. **C** Classifier embedding in CamoTSS includes a logistic regression model and a convolutional neural network model. Ranked ATAC-seq peaks were used as ground truth labels for the TSS clusters when training classifiers.

intron region), while the signal of H3K36me3 was mainly enriched downstream of TSS clusters detected by CamoTSS (Fig. 2E). We also checked the negative clusters (i.e intersect with negative scATAC-seq peak or without intersecting to any scATAC-seq peak), and found no signal surrounding these negative TSS clusters. As examples, Fig. 2F shows one canonical and one novel TSSs of DPH1 and two canonical

TSSs of SCP2 detected by CamoTSS (highlighted by red lines), all with signal support from histone modifications, RNA POL2 and scATAC-seq, which suggests high reliability of CamoTSS. Then we calculated the percentages of annotated and novel TSS clusters detected by CamoTSS in PBMC (Fig. 2G) and it shows the majority of identified TSS clusters (73.6%) are annotated. As we expected, most TSSs

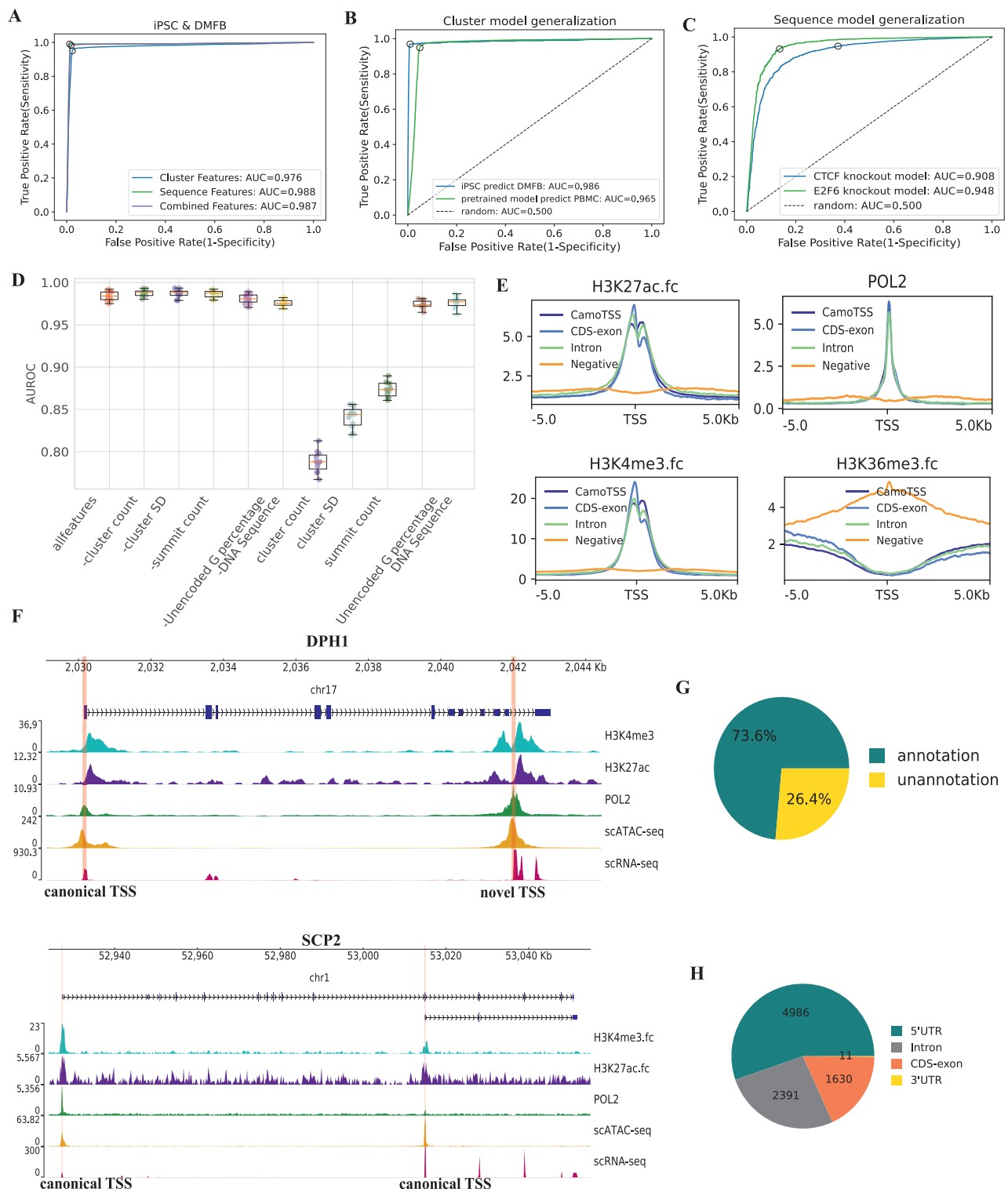

(4986 out of 9018) detected by CamoTSS were mapped to 5'UTR, while a substantial part of TSSs was also mapped to intron (2391) and exon (1630) regions, presumably due to the alternative usage of transcription start sites.

It is worth noting that users can customize the thresholds to filter TSS according to their preference when running CamoTSS. Here, we evaluated the minimum UMI and minimum inter-cluster distance based on the PBMC dataset to provide a reference. As depicted in Supplementary Fig. S3, both the cluster model and sequence model exhibit their decreased performance when the dataset is derived from

clusters with UMI counts <50 (but >10), which suggests that using > 50 UMI is more effective to filter false positive TSSs, while users may still use a lower threshold to detect rarer TSS if they can tolerate a lower sensitivity (the ROC curve shows our models can still achieve a good false positive control). For the parameter of minimum inter-cluster distance, it was used to control the distance of clusters when detecting the alternative TSS after they were retained by the classifier. Therefore, it does not largely affect the quality of clusters as shown in Supplementary Fig. S4C, D. On the other hand, a larger parameter of cluster distance can decrease the number of total TSS or genes with

**Fig. 2 | CamoTSS can accurately detect TSS. A** Receiver operating characteristic (ROC) curves for TSS classification with three groups of features by using logistic regressions; the curves are for pooled non-redundant TSSs form iPSC and DMFB datasets (Methods; individual sample shown in Supplementary Fig. S1). Ten-fold cross-validation is used for the evaluation. Source data are provided as a Source Data file. **B** ROC curves showing using iPSC dataset to predict DMFB dataset and using pre-trained cluster model (with combining iPSC and DMBF) to predict PBMC dataset with paired scATAC-seq and scRNA-seq data. Source data are provided as a Source Data file. **C** ROC curves for using samples which do not contain binding sites of the CTCF or E2F6 as training datasets to predict samples which only contain binding sites of the CTCF or E2F6. Source data are provided as a Source Data file. **D** All features (e.g. clusters features and sequence features), the combination

dropping one feature in all features and each one feature were fed to the logistic regression model to perform prediction. AUROC values are obtained via 10-fold cross-validation (*n* = 10). Source data are provided as a Source Data file. **E** The distributions of RNA POL2, H3K4me3, H3K27ac and H3K36me3 signals around the TSSs detected by us and the random regions produced by bedtools. RNA POL2, H3K4me3 and H3K27ac show enrichment around TSSs while H3K36me3 is enriched downstream of our TSSs. **F** Tracks plots of two examples (DPH1 and SCP2) show peaks of scRNA-seq, scATC-seq, POL2, H3K27ac and H3K4me3. Red lines denote the location of our detected TSSs. **G** Pie chart of the percentage of our detected TSS regions/clusters as annotated by reference genome or novel TSSs. **H** Genomic distribution of the detected TSS regions. Source data are provided as a Source Data file.

## CamoTSS facilitates cell identity analysis

Given that CamoTSS has the capability of detecting TSSs at a single-cell level, we wonder how much it can enhance cell identity analysis and to which extent it presents a cell-type specificity. Here, we downloaded 84,363 single-cell transcriptomes (profiled by 10x Genomics, 5' tagged scRNA-seq with reads 1) across 15 organs from one adult donor to detect the variability of TSS usage across cell types and organs[30]. By leveraging CamoTSS, we can obtain matched gene expression and TSS expression for each individual cell, allowing for direct comparison of RNA and TSS clusters without the need for integration methods. We took muscle (5,732 cells) as an example and used the same parameters to preprocess the gene and TSS expression matrices and clustered the cells (Supplementary Fig. S5). Overall, the majority of cells showed consistent clustering between using gene- or TSS-level expressions (Adjusted Rand index: 0.572; Supplementary Fig. S6). Interestingly, we noticed that NK/T cells have different clustering results when using TSS- or gene-level expressions as input (Fig. 3A). The NK/T cells were clustered as NK cells and T cells at gene level, while our TSS level analysis returns NK (& CD8 T) cells and CD4 T cells (Supplementary Fig. S7). While this difference in clustering preference mainly happens in the coarse resolution, it suggests the TSS-level contains complementary information for characterizing cells. We speculate this difference conveys that the extra information coming from the distinct promoter usage between cell populations, which links to the regulatory variability between cells that may be evident at the epigenetic level. To validate our hypothesis, we leveraged Single-Cell Regulatory Network Inference and Clustering (SCENIC)[31] to analyze gene expression data and obtain the AUCell score of regulons in each cell. This activity matrix was used to separately predict clusters of RNA level and TSS level with a logistic regression model. As we expected, the TSS clusters (S4 and S9) are more separable than the RNA clusters (R5 and R7) in this supervised approach (AUROC 0.982 vs 0.914; Fig. 3B), which is consistent visually in the top two principal components via an unsupervised manner (Supplementary Fig. S8), suggesting that the TSS expression profile contains regulation information.

From another perspective, the R7 cluster (identified at the gene level) exhibits two distinct TSS profiles (TSS clusters S4 and part of S9) that are highly concordant with the two subpopulations named CD8+ T cell (*GZMK* T cell) and CD4+ T cell (*IL7R* T cell) (Supplementary Fig. S7)[30]. We also detected the regulatory patterns of R7 cluster and found it displays different regulatory patterns between S4 and S9, for example, IKZF1, a regulator of lymphocyte differentiation (Supplementary Fig. S9A). Otherwise, the clear differential pattern of R7 disappeared when we shuffled the order of cells (Supplementary Fig. S9B), which suggests the TSS-level analysis can capture distinct regulation status that is masked by gene-level analysis. Furthermore, when comparing the transcription factors binding possibility to the top 1000 cluster-specific TSS regions by using Homer[32], we found that S4 and S9 showed enrichment of different transcription factor family

motifs, including nuclear receptors (NR) and basic leucine zipper (bZIP) respectively, as illustrated in Fig. 3C.

Subsequently, we surveyed the TSS profiles in all 15 organs and asked if the TSS profiles are more similar within cell types or organs. In total, 21,125 TSSs were detected by merging all cells from 15 organs and then used to calculate a Pearson's correlation coefficient between each cell type in each organ (at a pseudo-bulk level). By performing the hierarchical clustering on the correlation, the dendrogram shows that samples of the same cell type (across organs) have a higher similarity, with few exceptions (Fig. 3D), consistent with direct grouping by cell types or organs (Supplementary. Fig. S10). This "cell type-dominated clustering" pattern implies that most cell types possess a conserved TSS expression signature. To further illustrate the usage of the TSS profile at cell type identification, we compared the top 20 most significant markers at both TSS and gene levels for each cell type in the bladder (Fig. 3E). Venn diagrams show partial overlap between gene- and TSS-based markers in all cell types, indicating that TSS may serve as additional predictive features for cell type identification, for example, *RAMP3*_ENST00000242249 as a marker for endothelial cells and *C7*_ENST00000313164 for fibroblast (Fig. 3F, Supplementary Fig. S11).

Next, we focused on genes which have at least two TSSs. To identify cell-type differential expression at the TSS level, we performed a differential analysis between original annotated cell types and searched for genes with both cell-type-specific TSS shifts (one cell type vs each of others) and non-cell-type-specific TSS (one cell type vs any of others; Methods). We detected 2,301 genes containing such isoform markers in 15 organs (Supplementary Dataset S1, Supplementary Fig. S12). Figure 3G top panel shows an example of such TSS from the zinc finger E-box binding homeobox 2 (*ZEB2*) gene in muscle, which is known to be a transcription factor to regulate epithelial to mesenchymal transition associated with many cancers[33]. While the gene-level expression of ZEB2 is less distinct across cell types, we find that among the 2 TSSs detected by CamoTSS, one TSS without annotation (with minor expression), is almost exclusively expressed in satellite cells. The histone modification, RNA POL2 and scATAC-seq signals all appeared at the same position of TSS (Supplementary Fig. S13), which indicates the reliability of TSS identified by us. As another example, *MFSD1* plays an essential role in liver homeostasis as a lysosomal transporter[34]. It undergoes an isoform shift in fibroblast in the heart, where the expression of the ENST00000486568 is significantly higher, suggesting distinct cell-type-specific TSS localization (Fig. 3G bottom panel, Supplementary Fig. S13). To further determine the role of TSS as a cell type marker, we used all of these TSS-level cell-type markers (cell number > 50) to predict cell type annotated by gene expression profile from the original report, and achieved accurate predictions on all cell types in the esophagus (Fig. 3H). We were, then, curious about how many TSS markers of cell type are shared by other organs. To solve this problem, we used the upsetR package[35] to detect the intersection of genes including TSS markers among organs and found there are still some TSS markers overlapping across multiple organs (Supplementary Fig. S14). *SH3KBP1* shared the same TSS switch among 7 distinct organs in NK/T cell (Supplementary Fig. S15, Supplementary Fig. S16), indicating the

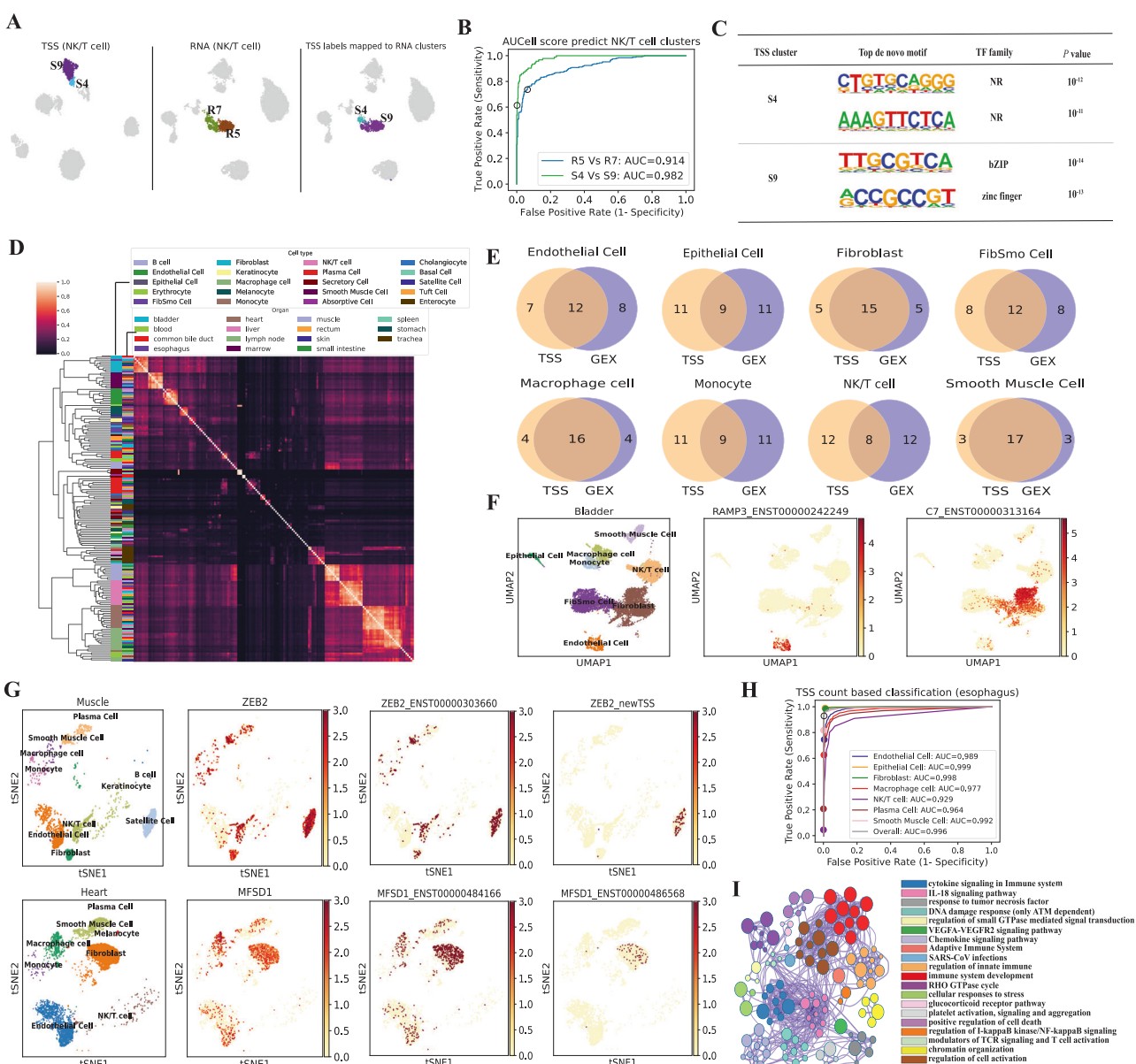

**Fig. 3 | CamoTSS analysis on TSSs between cell types across 15 human organs.**
**A** UMAP projection of TSS profile (left) and RNA profile (middle and right) in muscle. The T cell cluster is highlighted in the colors. All the other cells are colored gray. **B** ROC curves for NK/T cell clusters prediction by AUCell scores. Source data are provided as a Source Data file. **C** The top de novo motifs enriched in the top 500 cluster-specific peaks of S4 and S9. *P*-values were calculated by using binomial tests. **D** Heatmap of Pearson's correlation of expression of common TSS among all cells from all organs. **E** Venn diagrams of top 20 significant TSS markers and RNA expression markers in 8 cell clusters of the bladder. **F** UMAP plots of TSS data specific marker. **G** tSNE plots show alternative TSS marker masked at gene level.

**H** ROC curves for prediction of cell types from the first 20 PCs of TSS matrix by using a random forest in a multi-label classification. Models were evaluated by using 10-fold cross-validation, whereby the overall average is obtained by merging all cell types at a micro level. **I** Enrichment network representing the top 20 enriched terms of significant alternative TSS. The enriched terms that displayed high similarity were grouped together and presented as a network diagram. In this diagram, each node corresponds to an enriched term and is assigned a color based on its cluster. The size of each node reflects the number of enriched genes, while the thickness of the lines connecting nodes represents the similarity score between the enriched terms.

generalizability of TSS as a cell type marker. Next, we explore the biological function of the TSS markers of each cell type (Fig. 3I, Supplementary Fig. S17). Notably, the functions of signature TSS of NK/T cells are mainly enriched in terms related to various immune response processes, including immune system development, cellular responses to stress, regulation of cell activation and regulation of innate immune.

## Altered TSS usages in nasopharyngeal carcinoma microenvironment

Although alternative promoter usage has been found to be cancer type-specific and predictive of patient prognosis via bulk RNA-Seq[18], it

remains largely unexplored whether and to which extent the precise TSS usage at a single-cell resolution can further explain the heterogeneity in the cancer microenvironment. To explore this problem, we applied CamoTSS to a nasopharynx dataset (5' scRNA-seq, 10x Genomics) from patients with either nasopharyngeal carcinoma (NPC; *n* = 7 patients) or nasopharyngeal lymphatic hyperplasia (NLH; *n* = 3 patients), covering 51,001 cells in total (Fig. 4A, B), downloaded from a recent study[36]. Then we applied CamoTSS to the whole datasets and identified 7,058 TSSs, where 1,784 genes contained at least two TSSs. By employing BRIE2[25] on these multi-TSS genes, 547 genes were found with significant differential usage of alternative TSS between NPC and NLH (FDR < 0.01;

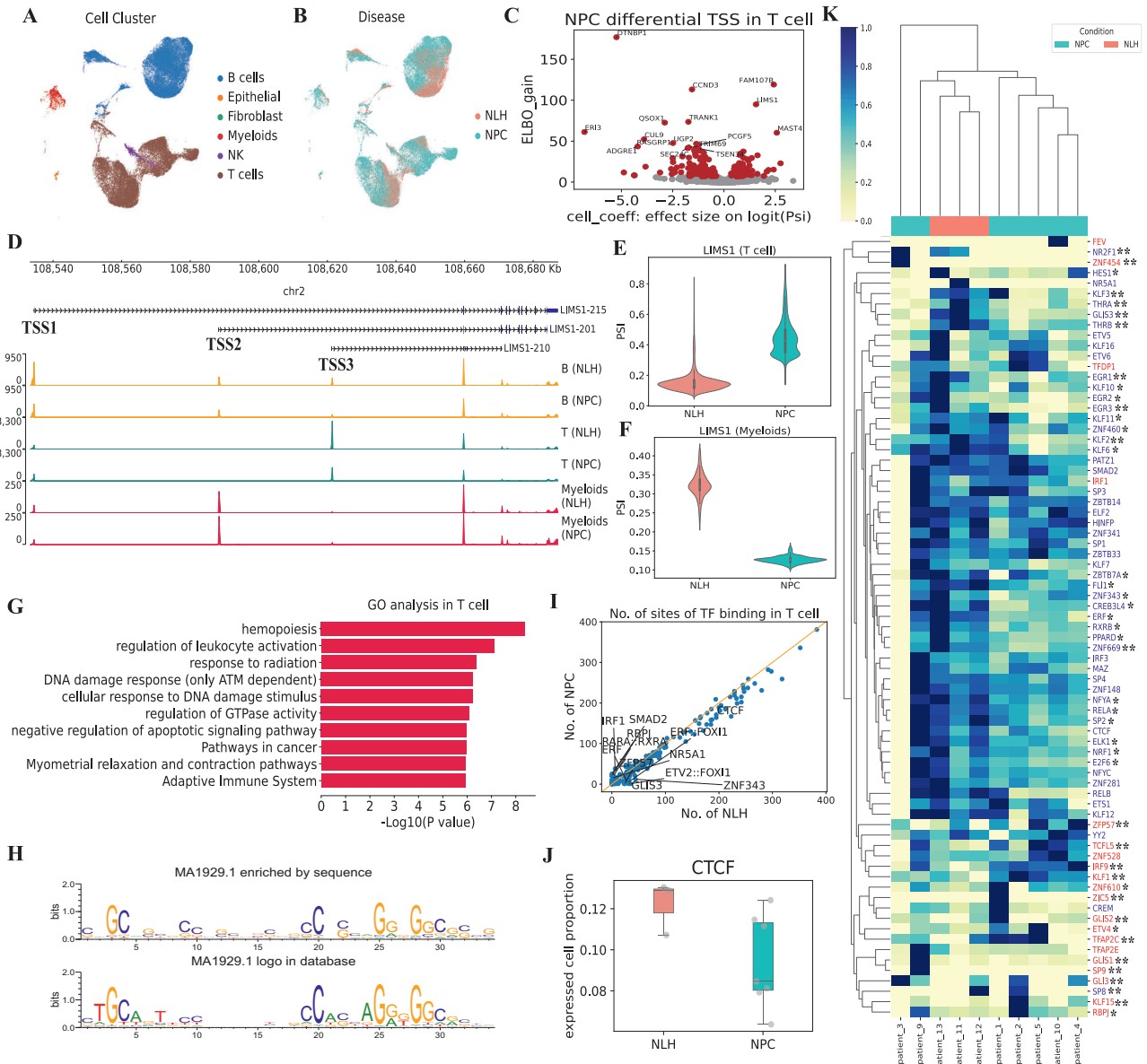

**Fig. 4 | CamoTSS identifies differential alternative TSS usage from nasopharyngeal carcinoma. A, B** UMAP plot of gene-level expression, annotated with cell types (**A**) and disease status (**B**). **C** Volcano plot to show the relationship between ELBO_gain and effect size on logit(PSI) for detecting differential TSS between NPC and NLH patients. Cell_coeff is the effect size on logit(PSI). Positive value means higher PSI in NPC. ELBO_gain denotes the evidence lower bound difference for the two hypotheses (Methods). **D** Genome track plot of LIMS1 in different cell types of NLH and NPC patients. One horizontal genome track denotes the coverage of all cells in one cell type. **E**, **F** Violin plot on example gene LIMS1 for T cell (E; $n$ = 6964 cells for NLH; $n$ = 17,607 cells for NPC) and Myeloids (F; $n$ = 158 cells for NLH; $n$ = 923 cells for NPC) in NLH and NPC patients. The y-axis PSI denotes the proportion of TSS1 (LIMS1-215; minor TSS here) among the top two TSSs in each cell type. **G** Bar plot showing the enriched terms of genes with differential TSS usage between NLH and NPC patients in the T cell. **H** WebLogo of the base frequency of MA1929.1 (i.e.

one motif of CTCF) enriched in the sequences detected by FIMO (top) and displayed in the JASPAR database (bottom). **I** Scatter plot of the binding frequency of human TFs on 528 TSS regions elevated in NLH and NPC patients (shown is based on T cells). Source data are provided as a Source Data file. **J** Box plot of expressed cell proportion of CTCF between NLH ($n$ = 3 patients) and NPC ($n$ = 7 patients). **K** Heatmap shows the hierarchical clustering of patients by the proportion of expressed cells of TFs that have significant differential binding frequency between NPC and NLH groups ($n$ = 10 patients). The color in the heatmap means the proportion of expressed cells with rescaling to the range of 0 and 1 on row. The up- and down- regulated TFs were displayed in red and blue, respectively (NPC vs NLH). Blue ID **: fold change < 0.6; Blue ID *: 0.6 < fold change < 0.8; red ID *: 1.2 < fold change < 1.5; red ID**: fold change > 1.5. Source data are provided as a Source Data file.

Supplementary Dataset S2). Take T cell as an example (Fig. 4C), among the genes that show alternative-TSS activation in NPC, multiple of them are well-known cancer-related biomarkers such as *QSOX1* serving as a prognosis biomarker in breast cancer[37], *DTNBP1* relating with memory and executive functions in brain tumor[38], and *FAM107B* associated with gastric cancer[39]. Particularly, two TSSs of *CCND3* have been reported to generate mRNAs with distinct 5' transcript leaders, resulting in protein isoforms with different N-termini[40] (Fig. 4C).

Additionally, cell type-specific differential TSS usages between cancer status were also identified, for example, *SIDT2* is only detected in B cells (Supplementary Fig. S18), which aligns well with a recent report on the cell-type-specific genetic effects to its isoform expression involving TSS changes[41]. Another prominent example is LIMS1, which contains three major cell-type specific TSSs, including TSS1 (*LIMS1-215*) as a leading TSS for B cells, TSS2 (*LIMS1-201*) for Myeloid cells and TSS3 (*LIMS1-210*) for T cells (Fig. 4D). Interestingly if focusing

on the proportion of the top two TSSs in T cells (TSS1 and TSS3), we further found that proportion of the minor TSS (TSS1, *LIMS1-215*) shows a significant up-regulation in cancer condition (NPC) compared to NHL (Fig. 4E), consistent with the trend of expressed cell proportions across patients (Supplementary Fig. S19SA, B). Surprisingly, Myeloid cells show an opposite trend, where the proportion of the same minor TSS (TSS1) decreases in NPC (Fig. 4F and Supplementary Fig. S19C, D), demonstrating the complexity of TSS regulation and its coupled modulation of cell types and disease conditions. To further understand the potential functions associated with NLH and NPC-specific TSSs at different cell types, we examined GO terms enriched in the TSS-shift gene sets between NLH and NPC for each cell type (Fig. 4G, Supplementary Fig. S20). Specifically, in T cells, the genes with differential TSS usage show enrichment in GO terms related to hemopoiesis, regulation of leukocyte activation, pathway in cancer and negative regulation of apoptotic signaling pathway, suggesting that an abundant TSS-mediated diversity is required for these genes associated with fundamental immune and cancer response properties.

To inform the potential regulatory mechanism leading to the alternative usage of cancer-related TSS, we used FIMO (v4.11.2) to find validated motifs in JASPAR in the NLH- and NPC-elevated TSS sequences. One prominent example is CTCF, whose protein level reduction or binding deficiency were found associated with the EBV-positive hypermethylation in NPC[42]. Here, by examining the JASPAR database, we found CTCF contains three binding motifs: MA0139.1, MA1929.1 and MA1930.1, which are highly similar to the sequence logo of its binding sequences detected by FIMO from our TSSs (Fig. 4H and Supplementary Fig. S21), confirming the reliability of our method. Then we counted the frequency of the database-curated motifs occurring in the two sets of TSSs ($n = 528$ for T cells and $n = 556$ for B cells) that are elevated in NLH or NPC groups. Interestingly, dozens of TFs have significant binding frequency changes between NPC VS NLH, favorably with down-regulation in both T cells (20 down-regulated vs 16 up-regulated, FDR < 0.05; Fig. 4I) and B cells (82 vs 19, FDR < 0.05 Supplementary Fig. S22), by large consistent with a recent report of EBV-positive NPC[42], suggesting a global change of transcription factor activities in this cancer. Such systematic bias is not likely caused by random chance, as no obvious difference is observed if randomly swapping the binding situation for NLH and NPC around 1000 times in T cells (Supplementary Fig. S23 left panel, K-S test: P-value = 9.09e-08) or B cells (Supplementary Fig. S23 right panel, K-S test: P-value = 3.2e-20).

To explore the underlying reason, the significant differential TFs were counted according to their classes. Nearly half (49.28%) of the differential TFs belong to C2H2 zinc finger factor class, while this proportion is only one-fifth when counting all TFs (Supplementary Fig. S24). All of these indicate the C2H2 zinc finger factor class play an essential role in the regulation of alternative TSS in this cancer, which has been reported in a previous study[43]. In addition, we also compared motif patterns bound by the top 10 significant TF with random motifs and discovered top 10 motifs have higher GC percentages, which is consistent with the binding pattern of C2H2 zinc finger factor class (Supplementary Fig. S25).

On top of that, to explain the inclined trend between NLH and NPC, we extracted 200bp (+/- 100 bp) sequence around significant TSS start of NLH and NPC separately as two groups test data and exploit our pre-trained convolutional neural network to predict the probability of being a positive sample. As Supplementary Fig. S26 shows, more TSSs in the NLH group were predicted as positive (i.e. the probability is >0.9) compared with the NPC group ($n = 239$ vs 175) and fewer TSSs in the NLH group were predicted as negative sample (i.e. the probability is <0.1) comparing with NPC group ($n = 175$ vs 229). The difference is significant ($P = 4.9$e-5, Fisher exact test), which indicates the sequence strengths around NPC-specific TSSs become weaker compared to NLH patients. We next asked if the expression profile of these TFs aligns with their binding frequency. Taking the critical and

well-studied CTCF as an example, though the expression level of expressed cells is similar between these two groups, the proportion of expressed cells is higher in NLH (Fig. 4J, Supplementary Fig. S27, fold change = 1.235), consistent with the report from another group[42]. Also, the whole proportion of expressed cell profiles of all significant TFs show a consistent trend with the binding frequency changes by large, and an unbiased clustering analysis of them well segregated NLH and NPC samples (Fig. 4K). Of note, the signal of TF expression is generally weak and our TSS analyses may further strengthen it via the binding site-based regulon activities.

## Alternative TSS usage in the tumor cells of gastric cancer

Besides the immune cells, we further studied the TSS switch in tumor cells compared to normal epithelial cells by analysing a recently published dataset on gastric cancer[44]. Specifically, we used the CamoTSS to obtain the transcription start site profile of 8,485 epithelial cells of tumor tissues (5,977 cells) and matched adjacent normal tissues (2,508 cells) from six primary gastric cancer (GC) patients (Fig. 5A). Next, 1,323 genes with multiple TSSs were selected and input to the BRIE2 to identify differential TSS usage between normal and tumor epithelial cells. As shown in Fig. 5B, 453 genes present significant TSS shifts (FDR < 0.01; Supplementary Dataset S3) between normal and tumor epithelial cells. As an example, *SLC29A1* (*hENT1*) shows a substantially decreased proportion of the upstream TSS (i.e., PSI value; Fig. 5C, D), due to the elevated expression of the shorter transcript ENST00000472176 (Fig. 5D), which may explain the finding that *SLC29A1* has a significantly higher expression level in gastric tumor tissue compared with normal stomach tissue[45]. Then we performed GO enrichment analysis to investigate which biological functions are associated with the alternative TSS usage events (Fig. 5E). We found that these TSS-shifting genes play an important role in Rho GTPases signaling which has been reported involved in most cancers[46]. Also, we further examined if these normal or tumor cell-preferred TSSs are enriched in any transcription factor binding motifs, again by counting the binding frequency with FIMO[47] (Fig. 5F; Methods) and testing the significance by Fisher exact test (Fig. 5F; FDR < 0.001 displayed here). 13 and 17 TFs were found significantly more frequent in normal and tumor preferred TSSs, respectively (FDR < 0.01; Supplementary Dataset S4). Multiple of them are involved in cancer-related regulation such as EBF1 modulating TERT expression in gastric cancer[48] and E2F8 exhibiting tumor-suppressing activity[49].

## Transcription start site shifting during human thymic development

Even though scRNA-seq is extensively used to study human development, the actual mechanism of choosing TSS in different development stages in various cell types remains unknown. To explore developmental-stage-specific TSS usage, we analysed a profile of transcription initiation events in three development time points of human thymus, including 11-week and 12-week in prenatal and 30-month in postnatal at distinct cell types (Fig. 6A, B). Next, we focused on genes with multiple TSSs and detected their TSS shift across time points at the single cell level respectively between 11-week vs 12-week, 11-week vs 30-month and 12-week vs 30-month (FDR < 0.01; Fig. 6C, Supplementary Dataset S5). Pairwise comparisons between time points show that similar genes with differential TSS-shifting patterns appear in 11-week Vs 30-month and 12-week Vs 30-month in T cells, which suggests that the closer the development distance, the more similar alternative TSS usage patterns are. The same trend was also observed in other cell types (Supplementary Fig. S28, Supplementary Dataset S5). We then summarized genes which are widely significantly differential in all three time points (Fig. 6D, Supplementary Dataset S6) or just have TSS-shift in a certain development time point (Supplementary Fig. S29, Fig. S30, Supplementary Dataset S6). For those triple differential TSSs (across all time point pairs), the dynamics patterns

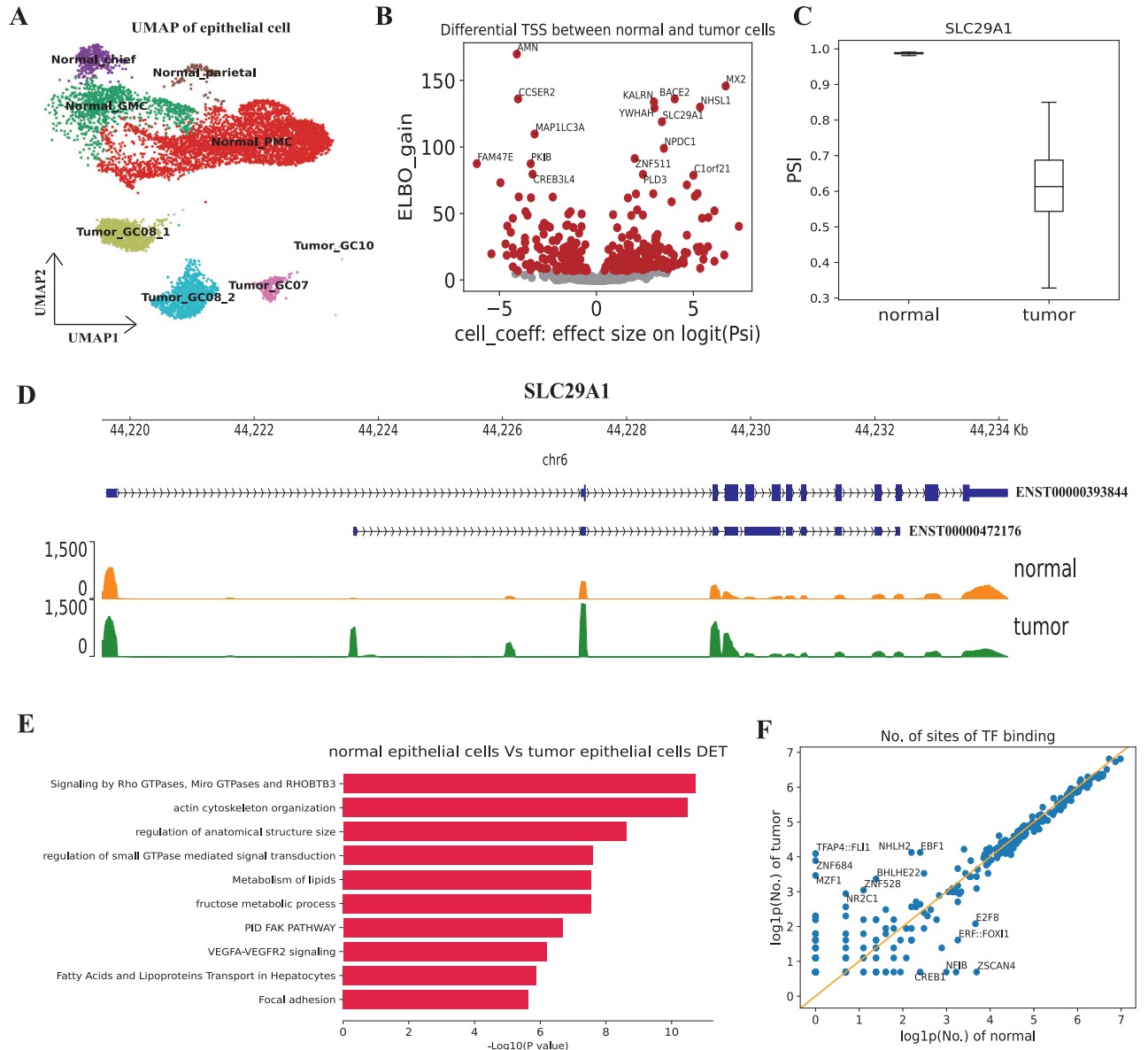

**Fig. 5 | CamoTSS detected transcription start site switch from epithelial cells of gastric cancer. A** UMAP visualization of epithelial cells (*n* = 8485) in gastric cancer. Each dot represents an individual cell, where colors indicate subcell type. **B** Volcano plot displaying the relationship between ELBO_gain and effect size on logit(PSI) for detecting differential TSS between normal and tumor cells. Cell_coeff is the effect size on logit(PSI). A positive value means higher PSI in normal cells. ELBO_gain denotes the evidence lower bound difference for the two hypotheses (Methods). **C** Boxplot showing an example gene that has a significant TSS usage between

normal (*n* = 5977) and tumor cells (*n* = 2508). **D** Genome track plot of *SLC29A1* in normal and tumor cells. One horizontal genome track represents the coverage of all cells within one specific cell type. **E** Bar plot exhibiting the enriched terms of genes with differential TSS usage between normal and tumor epithelial cells in gastric cancer. Source data are provided as a Source Data file. **F** Scatter plot of the binding frequency of human TFs from JASPAR database on 453 TSS regions elevated in normal and tumor epithelial cells. Source data are provided as a Source Data file.

can be either monotonic (38 for TSS1 increasing and 26 for TSS1 decreasing) or transient (17 for TSS1 upregulation first and 50 for TSS1 downregulation first; Fig. 6D, Supplementary Dataset S7); TSS1 denotes the most upstream TSS along the transcription direction. Notably, most genes containing TSS shift during three stages are well studied in immune development and promoter research such as *RAC2*, playing dual roles in neutrophil motility and active retention in zebrafish hematopoietic tissue[50] and *SLC3A2*, helping branched-chain amino acids (BCAAs) to control Regulatory T cell maintenance[51]. In addition, many genes with differential TSS usage only between two certain stages also play critical roles in development such as *FCHO1*, involved in T-cell development and function in humans[52] and *ST6GAL1* which can enhance B cell development and produces IgG in a CD22-

dependent manner in vivo[53]. To decipher the function of genes with alternative TSS, we performed GO enrichment analysis and found the enriched GO terms are highly relevant to cell cycle, development and stress response (Fig. 6E).

Next, inspired by the work from Haberle and colleagues[16], we also aimed to discover narrow shifts within one TSS region (i.e., cluster) during thymic development. The bona fide TSS clusters detected in the first step were selected and then we utilized a sliding-window approach to denoise the data and obtain reliable "CTSS" (CAGE-based TSS, i.e., at a single-nucleotide resolution) in one TSS cluster (Fig. 6F; Methods). Here, we applied the percentage of annotated TSS (allowing up- and down- stream 5bp shift) as evaluation criteria for the window size in PBMC dataset (mentioned in Fig. 2) and found that the window size

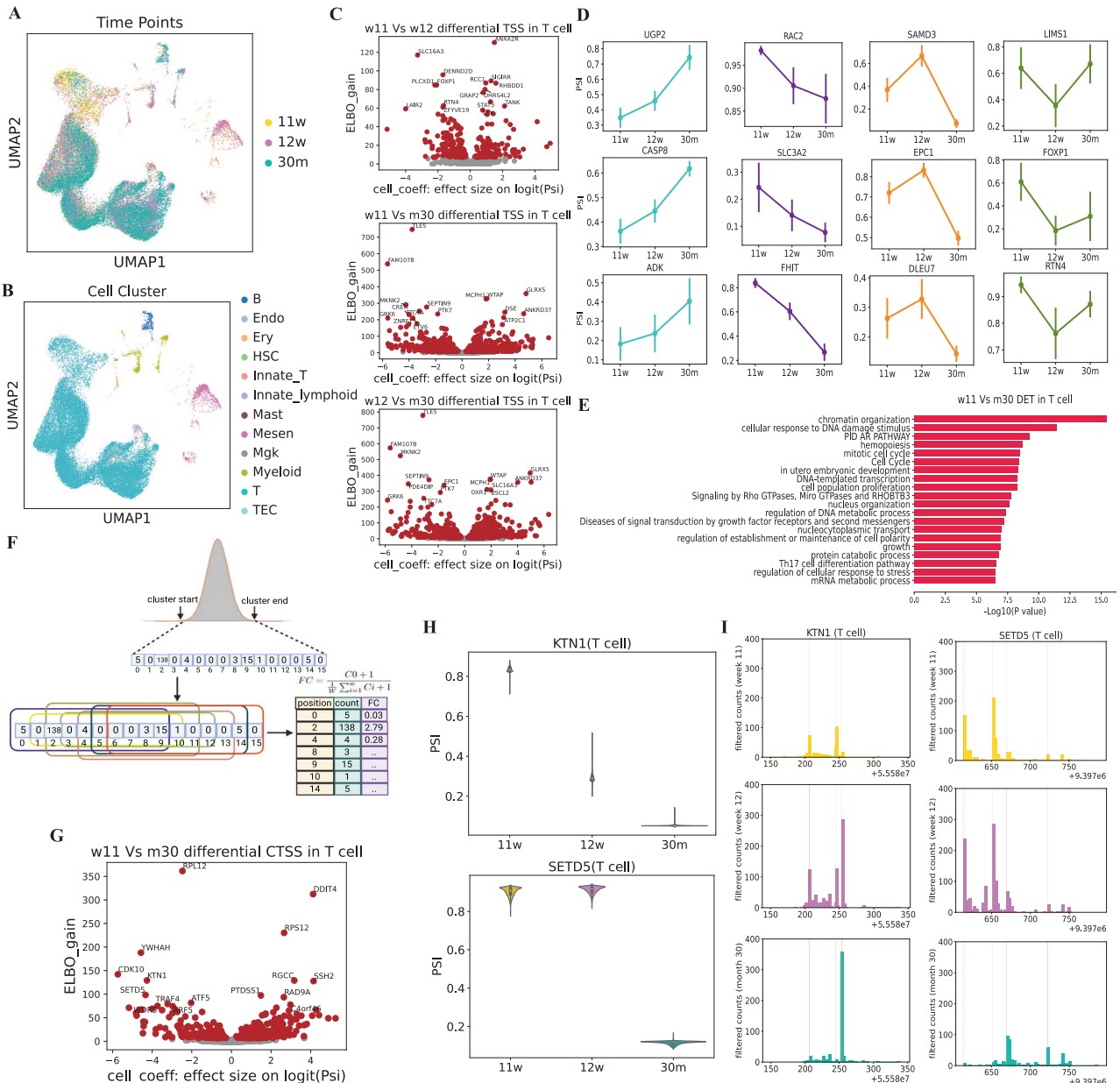

**Fig. 6 | CamoTSS identifies differential alternative TSS usage from human thymus development. A**, **B** UMAP visualization of all cells (35,629) in thymus development. Each dot is one cell, with colors coded according to the time points (**A**) and cell types (**B**). **C** Volcano plot showing the relationship between ELBO_gain and effect size on logit(PSI) for detecting differential TSS between week11 and week12 (Top), week11 and month30 (Middle), week12 and month30 (Bottom). Cell_coeff is the effect size on logit(PSI). Positive value means higher PSI in week12 (Top), month30 (Middle) and month30 (Bottom), respectively. ELBO_gain denotes the evidence lower bound difference for the two hypotheses (Methods). **D** Line chart showing four patterns of example genes which are all significant at three development stage pairs. Data are presented as mean values ± SD (n = 7059 cells for week11; n = 12,249 cells for week12; n = 13854 cells for month30). **E** Bar plot showing the enriched GO terms of genes with alternative TSS usage between week11 and

month30 in the T cell. Source data are provided as a Source Data file. **F** Illustration of the window sliding algorithm for identifying CTSS within one TSS cluster. Count and fold change parameters were used to filter noise. **G** Volcano plot between ELBO_gain and effect size on logit(PSI) for detecting differential CTSS between week11 and month30 in T cell. Same figure form as panel (**C**). Source data are provided as a Source Data file. **H** Violin plot of PSI value of KTN1 and SETD5 among week11 (n = 7059 cells), week12 (n = 12,249 cells) and month30 (n = 13854 cells). The two farthest CTSSs were picked up to calculate PSI for each gene. **I** Histogram showing the coverage of reads 1 with unencoded G at the cap obtained from 5' scRNA-seq in *KTN1* (Left) and *SETD5* (Right). The gray and red lines represent CTSSs identified by CamoTSS, while the red line shows the two farthest CTSS used for differential CTSS analysis with BRIE2[25].

does not have too much impact on the numbers and accuracy detected CTSS (Supplementary Fig. S31). To detect these narrow shifts within one TSS cluster (usually within 100bp), we first selected two farthest CTSSs within one cluster and then exploited BIRE2[25] to detect alternative CTSS usage during the three time points in each cell type. In total, we found 1891 genes with significant CTSS shifts between any two development points (233 between 11-week and 12-week, 663

between 11-week vs 30-month, and 995 between 12-week vs 30-month; FDR < 0.01; Fig. 6G and Supplementary Fig. S32), with higher overlap between the two prenatal stages vs the postnatal stage. In other words, these significant TSS regions contain CTSS shifting from one end to another during thymus development, for example, *KTN1* and *SETD5* (Fig. 6H–I, Supplementary Fig. S33), both of which were reported as important genes for development in zebrafish[54] or mammals[55]. To

unveil the potential functions of these genes with narrow shifting CTSSs between each pair of time points, we performed GO terms enrichment and found that recurrent terms included cell cycle, chromosome organization and translation (Supplementary Fig. S34), suggesting that the alternative CTSS usage within one TSS region may play an essential role in thymus development.

## Discussion

In this work, we present CamoTSS, a computational method for de novo detecting TSS from 5' tag-based scRNA-seq data. This method enjoys a data-driven design and embeds a classifier to accurately detect TSSs, and enables efficient identification of alternative TSS usage between single-cell populations by seamlessly leveraging our BRIE2 model. Specifically, this method first adopts a hierarchical clustering algorithm to determine the potential TSS clusters, followed by filtering the false positive clusters mainly via an embedded classifier from reads and/or sequence-based features. Finally, it annotates those genuine TSS clusters with known transcript annotations (e.g., from a GENCODE GTF file) by a Hungarian algorithm and counts the UMIs of each TSS at a single-cell level. Generally, the four reads-based features with a logistic regression model provide high accuracy, hence are used in practice by default. On the other hand, our convolutional neural network module, by extracting sequence features from the query TSS, can achieve comparable performance and further improve the accuracy when combined with the reads-based features. Additionally, as shown in the NPC data, this sequence-based model can further reveal the weakened TSS patterns in a disease condition.

While CamoTSS can identify all TSS clusters, we focused on the analysis of alternative TSS usage in this study, covering broad biological scenarios including cell types across 15 human organs, cancer conditions from multiple samples and thymus development with three time points. Differential TSS usage in different cell types was observed in multiple organs, where TSS clusters provide additional molecular signatures otherwise masked by gene expression and help to identify cell clusters with higher purity. Importantly, compared to NLH individuals, differential TSS usage, especially a general preference toward weaker promoter, was detected in all major nasopharynx cell types of NPC patients, which may be regulated by TFs from C2H2 zinc finger factor class. In addition, we also found hundreds of genes with TSS-shift during thymus development stages and many of them have narrow shifts within 100bp in a cell type-specific manner. Taken together, the TSS-level information, especially the alternative TSS usage, can provide more detailed cellular phenotypes and may imply regulatory patterns across various biological contexts. Considering the almost free access to the TSS from the 5' scRNA-seq data, CamoTSS may introduce a new paradigm in analysing such data and resolving the cellular heterogeneity in a finer-grained resolution with better interpretation from the regulatory perspective.

Additionally, there are also open challenges in the TSS analysis. First, when we performed alternative TSS usage, we mainly support the analysis of proportional change of the two major TSSs among cell populations. However, there can be more than two TSSs playing critical roles, especially when ranging from different cell types and diseases in one setting. Therefore, an extended model with support to jointly analyse multiple TSSs would account for such compositional change. Second, a large fraction of public 5' scRNA-seq datasets, e.g., on GEO, may only contain cDNA on read 2 for various reasons. Therefore, extending our CamoTSS to support a read2-only mode with high accuracy can be another timely contribution to data mining from the exponentially increasing single-cell data across the community. For addressing this potential challenge, our sequence-based neural network model would play a more important role in specifying the TSS region. Third, considering that the cell-by-TSS UMI count matrix can serve as a more informative input compared to the conventional cell-by-gene count matrix, it remains to be further examined if the downstream analysis pipeline needs to be adapted, especially when integrating with data from other platforms, e.g., 3' scRNA-seq data. Last, we also anticipate rapid advances in long-read technology for potential TSS analysis and CamoTSS may be further extended to support it.

## Methods

### scRNA-seq initial data analysis

The raw fasta files including reads1 (>100 bp) and reads2 were downloaded, and then sequences were aligned to the *Homo sapiens* reference genome (hg38) to generate pair-end read alignment bam file by using the cellranger count pipeline (with parameter –chemistry SC5P-PE) of 10x Genomics CellRanger (v3.1) software. The possorted_genome_bam.bam file was manually filtered by using xf:i:25 tag before performing reads counting, same strategy according to the 10x Genomics criteria[56]. In most instances, we aggregated the bam file from each donor sample by using an in-house script (https://github.com/StatBiomed/CamoTSS). In brief, it adds sample ID to the cell barcode by using *pysam* package[57] and then merges all bam files by using *samtools merge*[57].

### Construction of CamoTSS method

CamoTSS is a stepwise computational method to identify TSS clusters and it includes three major steps: to cluster reads into TSS clusters/regions, to detect true clusters, and to annotate bona fide TSS clusters.

**Step1: Cluster for reads start site**. Preprocessed bam file was used as input to perform clustering for subsequent promoter evaluation. We fetch all reads 1 for each gene by using BRIE[6]. All obtained reads 1 then filtered according to the cell barcode list specified by users. We remove strand invasion artifacts by aligning the DNA sequence starting from the -14 base and ending at the aligned start sit of reads1 to the TSO sequence (5'-TTTCTTATATGGG-3'). The read is regarded as a strand invader when the edit distance is <3. The edit distance was calculated by utilizing *editdistance* python package. The reads were also filtered according to the *SCAFE* criteria to precisely calculate the number of unencoded G[24]. In brief, we require reads 1 that (1) should contain the last 5nt of TS oligo (i.e. ATGGG) and the edit distance is <4, (2) start with a softclip region (i.e."S" in CIGAR string) and the value of "S" is >6 and <20, (3) the match region following the softclip region is >5 bp. If the number of fetched reads is >10,000, we randomly selected 10,000 reads from all reads in that gene to make sure the efficiency of our software.

The start position of reads was extracted and then input to agglomerative clustering which is a kind of from-bottom-to-up hierarchical clustering method. In brief, it first determines the proximity matrix containing the Euclidean distance between each start position using a distance function. Then, this matrix is updated to display the distance between each cluster. Here, we use the average linkage method to define the distance between two clusters, which can be calculated by (1).

$$L(A,B) = \frac{1}{n_A n_B} \sum_{i=1}^{n_A} \sum_{j=1}^{n_B} D(X_{A_i}, X_{B_j}). \tag{1}$$

If the linkage distance between two clusters is at or above 100bp, then the two clusters will not be merged.

**Step2: Filter false positive cluster**. Although we discarded some aberrant reads which are strand invasion artifacts and affect the counting of unencoded G, quite a lot of clusters still can be observed located at the end of the gene body via integrative genomics viewer (IGV). We added another filtering step to efficiently identify high-confidence transcription start site clusters, by utilizing the power of ATAC-seq data given that TSSs are generally tagged as open regions. Specifically, for our main ATAC-seq data sets (DMFB and iPSC), the

bigwig file was downloaded and transformed to a bedGraph file by using UCSC Genome Browser's *bigWigToBedGraph* tool. Then *liftOver* was utilized to convert the bedGraph coordinates from hg19 to hg38. Peaks from ATAC-seq were ranked based on the *p*-value and the top and bottom 5% peaks were defined as ground-truth positive and negative peaks, respectively. Then the bam files of DMFB and iPSC from scRNA-seq were input to our software CamoTSS to detect clusters without filtering with a classifier. These clusters were defined to gold positive ($n = 5560$) and negative samples ($n = 5432$) by using bedtools (v2.26.0)[58] to intersect (parameter: -f 0.1) with ATAC-based positive and negative peaks, followed by removal of double-detected TSS regions if combining multiple datasets. Of note, the candidate TSSs without intersecting to any ATAC peak should also be taken as negative samples when using lowly covered data, e.g., scATAC-seq for our PBMC data. Last, we used a subsampling method to ensure a balanced training set. Then, we designed and tested the following three models to distinguish high-confidence TSS from false positive clusters, and compared them with the above-annotated data to select the most suitable model to embed into our pipeline.

1. The first model is "logistic regression" with four reads-based features (next paragraph). Specifically, we extracted four features including cluster count, summit count, the standard deviation, and unencoded G percentage for each cluster. The cluster count refers to the total UMI counts within one cluster and the summit count is the maximum UMI count for a certain position within the cluster. The standard deviation was calculated by *statistics* python package to measure the dispersion of the cluster (treating each UMI as a sample). An intact mRNA with the cap structure can reverse transcribe to cDNA possessing an additional dGMP and cDNA with an extra dGMP cannot be produced by cap-free RNA[59]. This evidence suggests unencoded G percentage within one cluster can be regarded as an essential feature to identify the transcription start site. Because of the number uncertainty of extra added dGMP, reads whose CIGAR string starts with "14S", "15S" and "16S" are all considered as reads with unencoded G. Finally, total samples ($n= 10,992$) with four properties were used to train logistic regression model at 10-fold cross-validation by implementing *Scikit-learn* python package (v1.0.2). We choose 0.5 as the default threshold and classify samples with a probability >0.5 as a true transcription start site cluster.

2. The second model is a "convolutional neural network". The same dataset ($n = 10,992$) used in the logistic regression model was also used to train this deep learning model. We utilized bedtools (v2.26.0) : getfasta to extract 200bp sequence shifting around the cluster summit position. These sequences were transformed to numbers between 0 and 3 representing the 4 possible nucleotides and were then one-hot encoded to provide a categorical representation of nucleotide in numerical space to train this neural network (i.e. A: [1,0,0,0], T: [0,1,0,0], C: [0,0,1,0], G: [0,0,0,1]). The convolutional neural network was constructed with PyTorch (v1.12.1) and it consists of two convolution layers connecting to Rectified Linear Unit (ReLU) for activation, followed by batch normalization, one max pooling and a dropout layer (probability of dropout: 0.4). The output from dropout layer was flattened and fed to fully connected layers (32 neurons) with a ReLU activation function. Then the second fully connected layers of 2 neurons were connected with the sigmoid function to calculate the probability of classification. The first convolutional layer has 128 filters with 8-mer width and 4 channels. The second convolutional layer has 64 filters, where the filter size is 4x4. This model can be summarized as follow,

$$\overline{O}_i = f^{Sigmoid} f^{Linear} f^{Linear\_ReLU} f^{Flatten} f^{Dropout} f^{Maxpooling}$$
$$f^{BatchNorm} f^{Conv\_ReLU} f^{conv\_ReLU}(\overline{X}_i) \quad (2)$$

where $\overline{X}_i$ denotes the one-hot encoded matrix (4, 200). All 10,992 One-Hot-Encoded matrices were split into a training set, test set and validation set according to the 6:2:2 ratio. Then this model was trained with a batch size of 256 and 500 epochs with SGD optimizer with a learning rate of 0.003 and momentum of 0.8. The model with the lowest validation loss during training was kept at last.

3. The third model is the "combination of logistic regression and convolutional neural network". The 32 dimensions features of the first fully connected layer were saved and combined with the four cluster features mentioned above as input features (36 dimensions) to the logistic regression. The same dataset was used to train logistic regression at 10-fold cross-validation. The difference between this model and the first model is the features used to train logistic regression changing from 4 to 36 (i.e., 4+32). Of note, we have also implemented another version of the combined model with jointly training the CNN models, namely concatenating the four reads-based features to the second-last layer before the sigmoid activation (namely it becomes 36 instead of 32). This setting archives minor improvement compared to the separated training, hence is only supported as an option to choose.

We assess the performance of the three methods above by plotting the receiver operating characteristics (ROC) curves and calculating the area under the curve (AUC) values, which can systematically evaluate the sensitivities and specificities of models.

Of note, we have also included threshold-based filtering before the classifier by using the total read counts within one cluster and after the classifier by the distance between two neighboring clusters. Both parameters can be flexibly adjusted according to specific preferences.

**Step3: Annotate clusters.** In order to connect detected clusters with existing gene annotation, the start site of transcripts from a comprehensive gene annotation GTF file was assigned to the position with the highest count as the label of clusters by using the Hungarian algorithm. The cost matrix was created by calculating the distance between the start site of each known transcript and the summit position of a query TSS cluster (i.e., with the highest UMI counts within each cluster). Our goal is to find a complete assignment of clusters to transcripts with overall minimal distance, which means to minimize Eq. (3),

$$X^* = \operatorname{argmin} \sum_{c=1}^{n_c} \sum_{t=1}^{n_t} C_{c,t} X_{c,t} \quad (3)$$

where X is a boolean matrix (X[c,t] = 1 if row c is assigned to column t) and C is the cost matrix, denoting the genomic coordinate distance mentioned above. The optimize.linear_sum_assignment function from *scipy* python package was used to solve this assignment problem. After assigning a transcript to one cluster, we named this cluster as the corresponding transcript if the transcript is in the cluster. On the other hand, the cluster is defined as a new TSS.

**Detect CTSS within one cluster.** Once the TSS clusters (regions) were detected, CamoTSS can further support detection of CTSS at a near single-nucleotide resolution. First, reads 1 were filtered to keep those with 'unencoded G' (i.e. CIGAR string starts with "14S", "15S" and "16S"). Then the UMI counts for each position were calculated. To further denoise the signal of CTSS, a sliding window algorithm was used to calculate the fold change, as follows,

$$FC = \frac{C_0 + 1}{\frac{1}{W}\sum_{i=1}^{W} C_i + 1} \quad (4)$$

where $W$ is the window size, and the default value is 15 in CamoTSS, and $C_i$ denotes the UMI counts for the $i$ position within the downstream-oriented window ($i = 0$ means the query position). The bona fide CTSSs

within each cluster were obtained after filtering according to the fold change and UMI counts values defined by users. In the thymus development dataset, we used the default setting (fold change = 6, UMI counts = 100).

**Analysis of scATAC-seq PBMC dataset.** The PBMC dataset with paired scRNA-seq and scATAC-seq was download from ArrayExpress (Data availabilty). The raw scATAC-seq data was processed with cellranger-atac (v2.1.0). Reads were aligned to hg38 reference genome download from 10x Genomics website. Then the MACS3[60] was applied to call peak [option -g hs -B -q 0.01]. Then the approach to acquiring positive samples remains unchanged from the aforementioned method. We selected the clusters intersected with low confidence scATAC-seq peaks and no scATAC-seq peak at all as negative samples.

### Evaluation of epigenetic features and RNA POL2 enrichment of detected TSS

The processed histone modification data (i.e. bigWig file) of PBMC were downloaded from the Roadmap project in ENCODE. The target of histone Chip-seq includes H3K27ac (accession: ENCFF067MDM), H3K4me3 (accession: ENCFF074XHZ) and H3K36me3 (accession: ENCFF953FFP). The aligned RNA POL2 data of PBMC obtained from Chip-seq targeting POLR2A was downloaded from ENCODE and the accession is ENCFF595NCO. The fold change (FC) data of each signal compared with the input signal was used for the histone modification data. The scRNA-seq data of PBMC was dealt with CamoTSS to obtain the region of TSS (i.e. bed file). Additionally, we used bedtools random to generate a random set of intervals in bed format as the negative control. Then the computeMatrix from deepTools was utilized to calculate the histone signal score of TSS and random region [options: computeMatrix reference-point –referencePoint TSS -b 5000 -a 5000]. The matrix generated by computeMatrix was input to plotProfile to create a profile for the score.

Genomic tracks were obtained with pyGenomeTracks (v 3.7). We downloaded aligned bam files of PBMC scATAC-seq from ArrayExpress (accession ID: E-MTAB-10382). For scRNA-seq data and scATAC-seq, we use bamCoverage from *deepTools* to convert the alignment file of reads (bam file) to the coverage track (bigWig file). The inputs to *pyGenomeTracks* are bigWig files of scRNA-seq, scATAC-seq, RNA POL2 and other histone markers. The interval of TSS (bed file) detected by CamoTSS was used for highlight and the GENCODE GTF file (hg38) only containing the needed transcript was used for annotation.

### Analysis of genomic feature of TSS

We inputted the hg38 annotation file to *gencode_env* package (https://github.com/saketkc/gencode_regions) to obtain the genomic interval of 5' UTR, 3' UTR, intron and exon and then selected the start site of the TSS clusters as the symbol of them to count genomic feature distribution of TSS.

### Identification of differential TSS and CTSS on cell type, disease and development stage

Preprocessing of data was done by scanpy (v 1.9.3)[61]. For the raw TSS h5ad files (containing cell-by-TSS) of all three datasets, we filter cells according to the expression h5ad file (containing cell-by-genes). The cell annotation information and UMAP or tSNE visualization coordinates from the expression h5ad file were mapped to cells in the TSS h5ad file. For the 15 organ dataset, we normalized each cell by total counts (target_sum=1e4) over all genes. For NPC and thymic dataset, the TSS UMI reads counts were divided by the total number of reads in the same cell and then multiple with 1e6 to normalize to counts per million (CPM). Then all of these count matrices were transformed with log1p.

For detection of cell-type specific TSS masked at the gene level in the 15 organ dataset, we first filter cell types whose cell number is <100. For the remaining cell clusters, a *t*-test was performed and the

difference in expression mean was calculated for each TSS between the cluster and its complement on the log1p count. We picked up TSS clusters that were significantly upregulated in the cluster relative to the complement of the cluster. In addition, the alternative TSS within the same gene cannot display the same pattern. In other words, if the alternative TSS was upregulated compared with the remaining cell clusters, then the degree of upregulation cannot be significant. All *t*-tests used a significance level of FDR < 0.01 (Bonferroni corrected).

For NPC and thymic dataset, BRIE2 (v 2.2.0)[25] was utilized to identify differential disease-associated or development-associated TSS or CTSS at single-cell resolution. We built an h5ad file for each cell type containing two layers for the expression of two alternative TSSs with the highest expression (or two alternative CTSSs with the farthest distance) of the corresponding gene. Of note, we selected the most upstream TSS/CTSS along the transcription direction as the TSS1. The file containing cell detection rate and cell state information for each cell type was also created to input to BRIE2 as a design matrix. Detecting differential TSS was performed using the brie-quant module for all pairwise comparisons [options: –batchSize 1000000 –minCell 10 –interceptMode gene –testBase full –LRTindex 0] and genes with differential TSS between two diseases or development states were defined as FDR < 0.01. The BRIE2 model leverages a statistic ELBO_gain (the difference between the full model and reduced model on Evidence Lower Bound) for model selection, which approximates the difference of expected log-likelihood and is related to the likelihood ratio test. The output coefficient indicates the effect size on the logit scale of the PSI value.

### SCENIC gene regulation network analysis

To examine the regulation activity of transcription factors, we applied the single-cell regulatory network inference and clustering (pySCENIC[62], v0.12.1) by using normalized expression matrices of muscle[30]. The pipeline comprises three main steps: infer gene regulatory network (GRN) based on coexpression patterns, predict regulon based on motif discovery (cisTarget) and quantify the predicted regulon activity with AUCell scores[62].

### Hierarchical clustering analysis

To investigate the similarity of TSS profiles across different organs and cell types, we first normalized the combined TSS profile of 15 organs (i.e. combined at bam file level and then run CamoTSS). Specifically, we divided the UMI counts for each TSS by the total UMI counts for all TSS in each cell type. The Pearson correlation coefficient was calculated by using normalized TSS expression of each cell type in each organ and then used to perform a hierarchical clustering analysis.

### Functional enrichment and motif enrichment analysis

We utilized the Metascape online web server (v3.5.20230101)[63] to perform GO enrichment analysis [options: Expression Analysis] and selected the top 20 enriched terms to do enrichment network visualization. Then we used Cytoscape (v 3.9.1) to modify and visualize the network of enriched terms.

For 15 organs dataset, we selected the top 1000 highly expressed TSSs for S4 (cells in R7) and S9 clusters and extracted sequence of the +500bp/-100bp around these TSSs respectively. Then findMotifs.pl in Homer (v4.11)[32] was applied for de novo motif discovery with responding the another sequence as background.

We used bedtools getfasta to extract the sequence of NPC- and NLH-elevated TSSs (proportional) and then downloaded 727 human TF motifs from JASPAR CORE 2022[64]. FIMO (v 4.11.2 MEME-suite)[47] was used to search TF motif occurrences within the TSS sequence of different conditions. Significant occurrences were defined by a q-value threshold of 0.05. Then we counted the occurrence frequency of each TF motif in the TSS sets in each disease state and calculated statistical significance by using Fisher's exact test: *p*-value < 0.01. Then each patient's average expression of these significant TFs was used to perform hierarchical

clustering. Weblogo (CLI)[65] was used to generate sequence logo of FIMO-searched and database-downloaded sequences [options: -F pdf -A dna –color-scheme classic –fineprint "" –errorbars No].

## Statistics and reproducibility

Unless explicitly stated otherwise, the central line, box boundaries, and whiskers of all box plots in this study represent the median, the first and third quartiles, and 1.5 times the interquartile range, respectively. Unless otherwise noted, p-values comparing distributions between groups across box or bar plots were calculated using unpaired two-sided Wilcoxon rank sum test, with Benjamini-Hochberg correction for multiple comparisons where appropriate.

We did not employ any statistical methods to predefine the sample size for analysis. Instead, we utilized all available samples as described and provided in the literature for each study. We ensure the reproducibility of the computational analysis in this manuscript through providing both the datasets and analysis code in the github and data availability, which is publicly accessible.

## Reporting summary

Further information on research design is available in the Nature Portfolio Reporting Summary linked to this article.

## Data availability

The 5' scRNA-seq and bulk ATAC-seq of iPSC and Human dermal fibroblasts used to do training dataset were downloaded from ArrayExpress under the accessions "E-MTAB-10385 [https://www.ebi.ac.uk/biostudies/arrayexpress/studies/E-MTAB-10385?accession=E-MTAB-10385]" and "E-MTAB-10381 [https://www.ebi.ac.uk/biostudies/arrayexpress/studies/E-MTAB-10381?accession=E-MTAB-10381]"[24]. The matched PBMC datasets used to check the performance of CamoTSS were also download from ArrayExpress under the accessions "E-MTAB-10378 [https://www.ebi.ac.uk/biostudies/arrayexpress/studies/E-MTAB-10378?accession=E-MTAB-10378]" (5' scRNA-seq) and "E-MTAB-10382 [https://www.ebi.ac.uk/biostudies/arrayexpress/studies/E-MTAB-10382?accession=E-MTAB-10382]" (scATAC-seq)[24]. Previously published 5' scRNA-seq data that were reanalyzed here are available in the GEO, ArrayExpress or GSA under the primary accession code "GSE159929 [https://www.ncbi.nlm.nih.gov/Traces/study/?acc=SRP292721&o=acc_s%3Aa]" (15 organs)[30], "GSE150825 [https://www.ncbi.nlm.nih.gov/Traces/study/?acc=SRP262300&o=acc_s%3Aa]" (naso-pharyngeal carcinoma)[36], "E-MTAB-8581 [https://www.ebi.ac.uk/biostudies/arrayexpress/studies/E-MTAB-8581?query=E-MTAB-8581]" (human thymic development)[66], "HRA000704 [https://ngdc.cncb.ac.cn/gsa-human/browse/HRA000704]" (gastric cancer)[44]. The cell type annotation is available within the article and its supplementary files, with a copy in the reproducibility GitHub repository (https://github.com/StatBiomed/CamoTSS). JASPAR database (2022) can be accessed by https://jaspar.genereg.net/. All data supporting the findings of the study are available within the article. Source data are provided with this paper.

## Code availability

CamoTSS is a publicly available Python package at https://github.com/StatBiomed/CamoTSS and https://doi.org/10.5281/zenodo.8343616[67]. Detailed documentation and analysis procedures to reproduce results in this paper are also uploaded to this repository.

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

## Acknowledgements

We thank Chen Qiao and Weizhong Zheng for technical help on CNN model building and troubleshooting. We thank Xianjie Huang for helping make the package efficient and for troubleshooting. We thank Jiaqi Li for

the suggestion on transcription factor detection. We thank Lanqi Gong, Shuai He and Runda Xu provided the annotation and UMAP/tSNE coordinate for NPC, 15 organ dataset and gastric cancer dataset. We also thank Lanqi Gong's discussion on TF trend analysis. This project is supported by the National Natural Science Foundation of China (No. 62222217), Innovation Technology Commission Funding (Health@InnoHK) and the University of Hong Kong through a startup fund and a seed fund (Y.H.). R.H. is supported by the Postgraduate Scholarship of the University of Hong Kong.

## Author contributions

Y.H. conceived and supervised this study. R.H. implemented the CamoTSS and performed all data analysis. C.C. provided guidance on CTSS and unencoded G analyses. R.H. and Y.H. wrote the manuscript.

## Competing interests

The authors declare no competing interests.
