## [Peer Review File · Nature Communications]

CamoTSS: analysis of alternative transcription start sites for cellular phenotypes and regulatory patterns from 5' scRNA-seq dataReviewer #1 (Remarks to the Author):

The manuscript presents CamoTSS, a computational method suite for identifying and quantifying transcription start sites (TSS) and analyzing their alternative usage from 5' scRNA-seq data. The authors illustrate the effectiveness of CamoTSS on multiple public datasets, including human cell atlas, primary nasopharyngeal carcinoma tissue, and human thymic cells in development and postnatal life. They demonstrate the method's ability to detect TSS with high specificity and reveal regulatory patterns.

Major:

1. The authors claimed that providing CamoTSS-identified TSS usage information enhances cell clustering results as compared to using gene expression level alone. For example, the R7 cluster (Figure 2A) obtained from gene expression-only clustering is impure, and the TSS-enhanced clusters S4 and S9 more correctly reflect the clusters in the original publication (He et al., 2020). However, the original publication also obtained the clusters solely with gene expression levels and how can the authors convince the reader that the impurity of the R7 cluster is not due to inappropriate clustering, e.g., not enough cluster resolution?
2. The authors discussed altered TSS usage (preferred usage of weakened promoters) in nasopharyngeal carcinoma (NPC) mainly in the immune cell types. However, in their analyses, they compared the immune cells in the NPC microenvironment to the nasopharyngeal lymphatic hyperplasia (NLH) patients who may not have immune cells with a clinically normal condition. In this way, the authors could only conclude that immune cells in NPC are using weaker promoters than in NLH patients, but not in normal individuals. In addition, though nasopharyngeal carcinoma is a solid tumor, the authors completely omitted the discussion of TSS abnormalities among the epithelial cell types.
3. Although the main algorithmic workflow of CamoTSS is discussed in detail, a lot of the analytic methods used in the paper are not been elaborated in the Methods section. For example, the BRIE2 model which CamoTSS is based on, the transcription factor analysis using single-cell regulatory network inference and clustering (SCENIC) (Aibar et al., 2017) and Homer (Heinz et al., 2010), and the definition of ELBO_gain and cell_coeff of the differential analysis in Figure 3C and Figure 4C, were not sufficiently introduced and explained.

Minor:

1. Line 112, add the citation to SCENIC.
2. Line 117, add the citation to Homer.
3. Line 225, the definition of TSS1 is unclear. Is TSS1 the most upstream TSS along the transcription direction or the genomic direction?
4. It is better to illustrate the structures and locations of the TSSs of the transcript isoforms for the examples presented in the paper, e.g., Figure 2G, 2H, and Figure 4H.
5. The axis labels of the figures are too tiny to read and therefore are not publication-ready.

References

- Aibar, S., González-Blas, C.B., Moerman, T., Huynh-Thu, V.A., Imrichova, H., Hulselmans, G., Rambow, F., Marine, J.C., Geurts, P., Aerts, J., et al. (2017). SCENIC: single-cell regulatory network inference and clustering. *Nat Methods* 14, 1083-1086.
- He, S., Wang, L.-H., Liu, Y., Li, Y.-Q., Chen, H.-T., Xu, J.-H., Peng, W., Lin, G.-W., Wei, P.-P., Li, B., et al. (2020). Single-cell transcriptome profiling of an adult human cell atlas of 15 major organs. *Genome biology* 21, 294.
- Heinz, S., Benner, C., Spann, N., Bertolino, E., Lin, Y.C., Laslo, P., Cheng, J.X., Murre, C., Singh, H., and Glass, C.K. (2010). Simple combinations of lineage-determining transcription factors prime cis-regulatory elements required for macrophage and B cell identities. *Molecular cell* 38, 576-589.

Reviewer #2 (Remarks to the Author):

Hou et al. presented CamoTSS, a computational tool that can identify and quantify potential transcriptional start sites (TSSs) from 5' tag-based scRNA-seq data. In addition, CamoTSS can detect differential usage of TSSs between conditions. While numerous methods have been developed for scRNA-seq data analysis, few are targeted toward splicing analysis. CamoTSS was specifically designed to address this important yet understudied problem. The authors conducted comprehensive analyses and demonstrated the power of CamoTSS in revealing novel cell

populations that are not identifiable through traditional gene expression-based clustering analysis. Overall, the paper is well-written, and the results are convincing. I have a few suggestions about the paper that hope to further improve its quality.

1. In the overview section, the second and third paragraphs appear to contain too many detailed results, which may make it difficult for readers to grasp the overall steps involved in CamoTSS. It may be helpful to condense these paragraphs and focus on presenting a concise summary of the steps involved in the tool. This will allow readers to better understand the general process before diving into the specifics. The detailed results can then be presented in the later part of the Results section, where readers can fully appreciate the power of CamoTSS.

2. Filtering out false positive TSSs is a critical step in scRNA-seq data analysis, but it can be challenging, particularly when paired scATAC-seq data is not available. While the authors have provided pre-trained models for filtering in some datasets, this approach may not be feasible for all datasets. It is important to discuss and provide practical guidance to users on what to do when paired scATAC-seq data are not available.

3. Since CamoTSS depends on clustering of TSS reads, can you comment on the ability of CamoTSS to detect rare TSSs? Can you also comment on the impact of thresholds used in the analysis?

4. It would also be interesting to see applications of CamoTSS in cancer data as splicing aberrations is common in cancer. Demonstration in cancer data will make CamoTSS even stronger.

5. The text in the figures appears to be too small, which can make it difficult for readers to comprehend the content. To improve the readability of the figures, the authors may consider reorganizing the different components within each figure and increasing the font size. This can help to make the content easier to understand and reduce the strain on readers' eyes.

Reviewer #3 (Remarks to the Author):

In this paper the authors developed the CamoTSS method suite which is used to identify and quantify transcript start sites using five-prime single-cell RNA-seq data. Their method uses features from read ends as well as genomic context in a linear regression model to determine if clusters of start sites represent a true TSS. The detected TSS's can then be quantified and analysed. This was applied to human nasopharyngeal cancer data and human thymic development data to show that TSS's can be used in clustering analysis and teasing out differences between cell types and developmental changes not seen at the gene level. Single-cell analysis has mainly been focused on gene level changes in expression and the increased precision of using TSS's will very likely provide interesting observations in both pre-existing and future single-cell data. The application of the method, whilst mainly being observational, does detail how this method can be used in future studies and shows great potential. As the main result of this paper is the methodology, I believe the impact could be greatly improved with more details on the labelling strategy as well as evaluation of the features and thresholds used. Care needs to be made in the presentation of the results as the reader is required to make several assumptions to understand the figures. This makes it hard for me as a reviewer to evaluate the claims in the text as I cannot be sure I have interpreted the figures correctly. The figures in the supplementary are well described. I was able to download and successfully run CamoTSS with the default parameters, but ran into problems using the CTSS mode on the test data. I have detailed these comments below.

Major Comments:

1. Both PacBio and ONT are planning on releasing single cell versions of their technology. The authors should comment on the applicability of their tool on this coming technology or what/how much would need to be done to allow it to be used.

2. Several of the figures and associated text leave a lot of room for assumption by the reader and it is likely I have misunderstood them, meaning I cannot fully verify the claims of the paper. Below is a list of areas where clarity can be improved which directly affect the claims. Smaller comments on the clarity of figures are under minor comments.

* The paper claims that the separation using TSS profiling is more consistent than the original paper however unless the reader finds the paper they can not see if this is more consistent or not. I suggest the authors include the original clustering so readers can make a direct comparison.

* Figure 2E supports the claim that samples across organs have high similarity. It would be good to see how the histogram looks without clustering when grouped by organs and cells as a comparison to see how much higher this clustering is.

* The paper claims that the expressed cell proportion of CTCF in NLH is significantly higher than NPC is significantly higher figure 3J, but no statistical test is shown.

* It is stated that all significant TFs show a consistent trend and that they segregate the NLH and NPC samples. However Figure 3K doesn't show a striking difference in the clustering with only KLF6 looking like the only TF with a major difference. If this TF was removed from the analysis would the samples still cluster together. If you were to take the mean of each TF grouped by patient condition (removing patient _3 as an outlier) how many would have a significant difference between them?

3. The authors developed a combination of a linear regression model with 4 features with a CNN for the genomic sequences. As a key part of the paper, I believe further depth into the different choices made, and how they affect the performance of the model will greatly improve the paper. Below are my suggestions and queries regarding these models.

* ATAC peaks are theoretically not always specifically at the TSS region including the mentioned CTCF binding sites. This could potentially result in clusters not from real TSSs being incorrectly labelled as a positive. The authors should comment on this and explain how much the impact of this loose labelling may impact the performance of their model.

* I noticed that the number of positive and negative labels were close to being balanced (5560 and 5432 labels respectively). It is good that the authors report this information. I would like to know if it is a coincidence that the labels are balanced or was a method used to ensure a balanced training set? If so this should be detailed in the methods section of the paper.

* It would be good to provide more details on the negative labelled clusters as they are currently an enigma but are an important part in training the model. How many of these were associated with a gene? For those that were annotated, where were they found along the gene body? Given that there are reads associated with negative clusters, this suggests transcriptional activity from where they are transcribed, why would they be associated with low ATAC peaks? As artifactual clusters are filtered out, what else could be causing the existence of these negative clusters (this would be good to explain in the introduction/discussion to detail the challenge of predicting TSS regions)?

* Figure 1E shows there is an enrichment of several other factors associated with TSS regions around their positive labels. This is evidence that many of their positive labels are correct. These graphs could also be broken down to show this distribution around TSSs detected in the CDS-exon and CDS-intron region (in response to comment a) and also negative labels (in response to comment c).

*The CNN used random regions as negative labelled sequences. Therefore I would assume the model is learning the binding sites for particular transcription factors in the sample. If this assumption is correct it would mean that the model would not perform well in samples from other datasets reliant on other binding sites, or in other species. If this assumption holds, either this limitation needs to be discussed or to claim it is generalizable, an evaluation on a broad range of data needs to be shown.

* Clusters were filtered out if the inter-cluster distance was less than 300bp, in the methodology, clusters were merged if they were within 100bp distance from each other. In the event two clusters occur 200bp apart, are they both discarded, or is only one discarded? Was there any evaluation on the selection of these thresholds and does it greatly impact the performance of CamoTSS if they are altered and how? Similarly for the 15bp window size for CamoTSS?

* It would be good to include discussion or evaluation of the importance of the four features (five

if including genomic sequence) used in the model. This can be done with many methods including but not limited to removing each feature and seeing how the performance of the model is affected. *The inclusion of single-nucleotide resolution predictions in a TSS cluster is a great addition (CTSS). Can the authors comment on the likely accuracy of these predictions or include evaluation of how precise these are.

4. I was unable to run CTSS mode with the test data provided, getting the following error message "FileNotFoundError: [Errno 2] No such file or directory: 'CamoTSS_CTSS/count/afterfiltered.csv'". If the test data is insufficient for this method it would be good for users to provide one.

Minor Comments:

1. When clusters are annotated to genes, is there a maximum distance threshold used? If so, this should be listed in the methods.

2. Below are comments related to the clarity of the figures and the associated text

* Figure 1B is hard to understand as many aspects of the figure are not labelled or described.

Does C stand for cluster? Are the lines within the circles meant to represent a string of alignments and the red circles highlighting a read start? This should be added in the legend.

* In the legend for Figure 2A, it should be made clearer that the 3rd UMAP is the clusters from the RNA clustering.

* Figure 2C mentions merged data. It should be made clearer if the merging was done in the original paper or in this paper.

* Figure 2E the heatmap dendrogram label columns are not labelled so it is not initially clear which column is cell types and which is organ.

* Figure 3D. TSS1 2 and 3 are not labelled on the figure, and the reader needs to look at a small transcript id to match them up. Additionally the vertical line hides the peaks making it harder to see differences.

* Figure 3D. Is this a genome track plot of one cell or the coverage across all cells of each type? Make this clearer in the figure legend.

* For readers that are less familiar with the biological background it would be good to include why CTCF is critical and well-studied and therefore chosen as an example.

* Figure 3E - Figure legend says this is for the LIST gene but the figure is labelled with LIMS. Make it clear if the violin plot contains the proportions of TSS1 from each cell.

* There is no description for the colour legend in figure 3K

3. When running CamoTSS with the default parameters it produced an empty CTSS directory. If no CTSS output is expected then it would be less confusing for users to not have this directory created.

4. The output fourFeature.csv does not have clear column descriptions. Either delete this output if it is an intermediate file not meant to be seen or add column names so that advanced users could use this information.

Reviewer 1: page 1-5
Reviewer 2: page 5-8
Reviewer 3: page 8-23
=====

Reviewer 1

The manuscript presents CamoTSS, a computational method suite for identifying and quantifying transcription start sites (TSS) and analyzing their alternative usage from 5' scRNA-seq data. The authors illustrate the effectiveness of CamoTSS on multiple public datasets, including human cell atlas, primary nasopharyngeal carcinoma tissue, and human thymic cells in development and postnatal life. They demonstrate the method's ability to detect TSS with high specificity and reveal regulatory patterns.

Response: Thank you for the summary and acknowledging our contributions.

Major:

1. The authors claimed that providing CamoTSS-identified TSS usage information enhances cell clustering results as compared to using gene expression level alone. For example, the R7 cluster (Figure 2A) obtained from gene expression-only clustering is impure, and the TSS-enhanced clusters S4 and S9 more correctly reflect the clusters in the original publication (He et al., 2020). However, the original publication also obtained the clusters solely with gene expression levels and how can the authors convince the reader that the impurity of the R7 cluster is not due to inappropriate clustering, e.g., not enough cluster resolution?

Response: We fully agreed with the reviewer on this. Indeed, when we increase the clustering resolution from our initial 1 to 4.5, both TSS and gene level analyses segregate the NK/T cell group into four clusters: T cell XCL1, T cell GZMK, T cell IL7R and T cell PRF1 (Fig. R1).

On the other hand, we want to clarify that our finding is that the largest difference happens between S4 and S9 at TSS level, while gene level detects the most distinct difference between R5 and R7. This suggests that the TSS level expression has complementary information to the gene level expression, presumably through gene regulatory patterns. To better demonstrate this, in this revision we further leveraged Single-Cell Regulatory Network Inference and Clustering (SCENIC)¹ to analyze gene expression data and obtain the AUCell score of regulons as a surrogate of the transcription factor (TF) activity in each cell. Impressively, we found that the activities of these 275 TF separate the S4 and S9 more clearly than that of R5 and R7, not only in a supervised way (via a logistic regression and 10-fold cross-validation; AUROC=0.982 vs 0.914, Fig. R2 and Fig. 3B in the revision) but also in an unsupervised manner via the top two principal components (Fig. R3 and Fig. S8 in the revision). These results

align well with our initial findings in R7 clusters where S4 and the S9 subset showed distinct activities on multiple TFs, e.g., IKZF1 (Supp. Fig. S9) and enrichment of different transcription binding motifs (Fig. 3C). Collectively, we hope the reviewer agrees with us that the TSS-level analysis is beneficial in characterizing cell states by prioritizing the regulatory information.

Fig. R1. UMAP projection of RNA profile (left panel) and TSS profile (middle and right panel) with increasing resolution.

Fig. R2. ROC curves for NK/T cell clusters prediction by AUCell scores of 275 TFs with a logistic regression model. Ten-fold cross-validation was used here.

Fig. R3. Scatter plot showing the top2 PCs when performing PCA for the AUCell scores. Dots were colored by RNA-level labels (left panel) and TSS-level labels (right panel), respectively.

2. The authors discussed altered TSS usage (preferred usage of weakened promoters) in nasopharyngeal carcinoma (NPC) mainly in the immune cell types. However, in their analyses, they compared the immune cells in the NPC microenvironment to the nasopharyngeal lymphatic hyperplasia (NLH) patients who may not have immune cells with a clinically normal condition. In this way, the authors could only conclude that immune cells in NPC are using weaker promoters than in NLH patients, but not in normal individuals. In addition, though nasopharyngeal carcinoma is a solid tumor, the authors completely omitted the discussion of TSS abnormalities among the epithelial cell types.

Response:

Thanks for pointing out this potential issue and we agree to further specify these. We changed the ‘cancer-specific TSSs’ to ‘NPC-specific TSSs’ and changed the ‘normal samples’ to ‘NLH patients’ (p.10).

Indeed, the analysis of the TSS abnormalities among the epithelial cell types is of high interest to us, too. However, due to the limitation of the dataset used in this paper which contains few epithelial cells (n=136 in total; 1-44 cells for each sample), we did not pay much attention to the epithelial cell. In addition, we also tried to get another NPC dataset that contains more epithelial cells ², but unfortunately, it was sequenced based on 3’ scRNA-seq.

3. Although the main algorithmic workflow of CamoTSS is discussed in detail, a lot of the analytic methods used in the paper are not been elaborated in the Methods section. For example, the BRIE2 model which CamoTSS is based on, the transcription factor analysis using single-cell regulatory network inference and clustering (SCENIC) (Aibar et al., 2017) and Homer (Heinz et al., 2010),

and the definition of ELBO_gain and cell_coeff of the differential analysis in Figure 3C and Figure 4C, were not sufficiently introduced and explained.

Response: Thanks for the comment. We now added the details of SCENIC and Homer analysis in the method *SCENIC gene regulation network analysis* and *Functional enrichment and motif enrichment analysis*, respectively. The description of the BRIE2 model has already been described in the method *Identification of differential TSS and CTSS on cell type, disease and development stage*, and we have further clarified it including the intuition of ELBO_gain.

The definition of ELBO_gain and cell_coeff were added to the legend of Figure 4C, Figure 5C and Methods.

Minor:

1. Line 112, add the citation to SCENIC.

Response: corrected; thanks.

2. Line 117, add the citation to Homer.

Response: corrected; thanks.

3. Line 225, the definition of TSS1 is unclear. Is TSS1 the most upstream TSS along the transcription direction or the genomic direction?

Response: Thanks for the great question. Actually, we did not introduce the definition of the TSS1 and TSS2 in the last version; it was more or less arbitrary. Here, according to your suggestion, we define the TSS1 as the most upstream TSS along the transcription direction. We have also updated all the TSS1 and TSS2 throughout the manuscript including Figure 4C, Figure 5C and Figure 5G. When we use BRIE2 to detect differential TSS usage between two conditions, it can only compare two TSSs for one gene. For TSS comparison in NPC and development datasets, we select the top two TSSs which have the highest expression values. Then the TSS1 was defined as the most upstream TSS. For CTSS comparison in the development dataset, we select the two TSSs which have the largest distance (i.e. the first TSS and the last TSS along the transcription direction). Also the TSS1 was defined as the most upstream TSS along the transcription direction. The clarification has been added to the result (line 262) and method (line 448) section.

Based on the new definition, Figure 5D and Supplementary table 5 have also updated accordingly.

4. It is better to illustrate the structures and locations of the TSSs of the transcript isoforms for the examples presented in the paper, e.g., Figure 2G, 2H, and Figure 4H.

Response: Thanks for the suggestion. The structures and locations of the TSSs for the examples presented in Figure 2G (Figure 3F in the revision), 2H (Figure 3G in the revision), and 4H (Figure 5H in the revision) were in the Supplementary Fig. S11, S13, and S33 in the revision. We have further highlighted them in the main texts now.

5. The axis labels of the figures are too tiny to read and therefore are not publication-ready.

Response: corrected; thanks!

Reviewer 2

Hou et al. presented CamoTSS, a computational tool that can identify and quantify potential transcriptional start sites (TSSs) from 5' tag-based scRNA-seq data. In addition, CamoTSS can detect differential usage of TSSs between conditions. While numerous methods have been developed for scRNA-seq data analysis, few are targeted toward splicing analysis. CamoTSS was specifically designed to address this important yet understudied problem. The authors conducted comprehensive analyses and demonstrated the power of CamoTSS in revealing novel cell populations that are not identifiable through traditional gene expression-based clustering analysis. Overall, the paper is well-written, and the results are convincing. I have a few suggestions about the paper that hope to further improve its quality.

Response: Thank you for the summary and appreciation of our work.

1. In the overview section, the second and third paragraphs appear to contain too many detailed results, which may make it difficult for readers to grasp the overall steps involved in CamoTSS. It may be helpful to condense these paragraphs and focus on presenting a concise summary of the steps involved in the tool. This will allow readers to better understand the general process before diving into the specifics. The detailed results can then be presented in the later part of the Results section, where readers can fully appreciate the power of CamoTSS.

Response: Thank you for your good suggestion. We took apart the overview section into two parts: "Overview of CamoTSS pipeline" and "Performance of CamoTSS in detecting TSS". The first part mainly focuses on presenting a concise summary of the steps involved in this tool. The second part presents the

performance of CamoTSS in detecting TSS.

2. Filtering out false positive TSSs is a critical step in scRNA-seq data analysis, but it can be challenging, particularly when paired scATAC-seq data is not available. While the authors have provided pre-trained models for filtering in some datasets, this approach may not be feasible for all datasets. It is important to discuss and provide practical guidance to users on what to do when paired scATAC-seq data are not available.

Response: Thanks for the comments - generalizability is indeed crucial. In our package, we have both cluster features (based on reads) and sequence features. The default pre-trained model is based on cluster features which we think are mostly technical related features (e.g., proportion of unencoded G). To demonstrate the generalization, we initially used iPSC as training to predict the DMFB, achieving high performance (AUROC=0.986, Fig. R4 & Fig. 2B in the revision). Here, to further support this in a more diverse cell population, we collected a published PBMC dataset with matched scRNA-seq and scATAC-seq data³, and used the pre-trained model (based on the combination of iPSC and DMFB) to predict the TSS candidates in this PBMC dataset. The prediction also reaches high overall performance (AUROC=0.965) and acceptable false positive rate (0.03), suggesting the good generalization of the pre-trained model (p.4).

Fig. R4 ROC curves showing using iPSC dataset to predict DMFB dataset and using pre-trained cluster model to predict PBMC dataset with paired scATAC-seq and scRNA-seq data.

3. Since CamoTSS depends on clustering of TSS reads, can you comment on

the ability of CamoTSS to detect rare TSSs? Can you also comment on the impact of thresholds used in the analysis?

Response: Thank you for your suggestion. From our understanding, rare TSS means low read counts in a cell population, either due to broadly low expression or low proportion of cells using it. In either case, we think the rare TSSs are more difficult to detect, because it is easier to confuse with the false positive TSSs. To assess this, we lower our filtering criterion from 50 UMIs to 10 UMIs and also group them into “10-50 UMIs” and “>50 UMIs” subgroups. Not surprisingly, we found that the AUROC dropped from 0.968 for “>50 UMIs” group to 0.908 for “10-50 UMIs” group, when using the cluster features (Fig. R5 and Fig. S3 in the revision). Of note, as the total UMI count is a feature in the model, we re-trained the model for each scenario here by a 10-fold cross validation. Interestingly, when using the sequence features, we found it is more robust to the rareness of TSSs (only a minor decrease in AUROC from 0.982 to 0.963). Another interesting phenomena is that even though the overall performance is weaker for rarer TSS (with 10-50 UMIs), it still can achieve a power of 0.75 with controlling the false positive rate < 0.01, with either cluster or sequence features. To conclude, we would recommend the users using our method to detect rare TSSs but remind them of the decreased sensitivity.

Fig. R5 ROC curves for different datasets splitted by cluster counts (i.e. 10 UMI and 50 UMI) and predicted by cluster model (left panel) and sequence model (right panel). Ten-fold cross-validation is used for the evaluation.

4. It would also be interesting to see applications of CamoTSS in cancer data as splicing aberrations is common in cancer. Demonstration in cancer data will make CamoTSS even stronger.

Response: Thank you for your comment. As you noticed, we have applied CamoTSS to the nasopharyngeal carcinoma (NPC) data, which is a squamous cell carcinoma arising from the nasopharynx epithelium. We guess that you

were suggesting performing analysis on the tumor cells (similar to the R1 point 3 on epithelial cells). Indeed, that is of great interest. Unfortunately, in this dataset, the epithelial cells are too few (n=136 in total; 1-44 for each sample), so we omitted it for further analysis. We also found another NPC dataset which includes a good number of epithelial cells², but the data was sequenced by 3' scRNA-seq.

Nevertheless, the TSS aberrations of the immune cells in the tumor microenvironment in NPC is also an important factor for cancer development and potential therapy.

5. The text in the figures appears to be too small, which can make it difficult for readers to comprehend the content. To improve the readability of the figures, the authors may consider reorganizing the different components within each figure and increasing the font size. This can help to make the content easier to understand and reduce the strain on readers' eyes.

Response: Corrected; Thanks!

Reviewer 3

In this paper the authors developed the CamoTSS method suite which is used to identify and quantify transcript start sites using five-prime single-cell RNA-seq data. Their method uses features from read ends as well as genomic context in a linear regression model to determine if clusters of start sites represent a true TSS. The detected TSS's can then be quantified and analysed. This was applied to human nasopharyngeal cancer data and human thymic development data to show that TSS's can be used in clustering analysis and teasing out differences between cell types and developmental changes not seen at the gene level. Single-cell analysis has mainly been focused on gene level changes in expression and the increased precision of using TSS's will very likely provide interesting observations in both pre-existing and future single-cell data. The application of the method, whilst mainly being observational, does detail how this method can be used in future studies and shows great potential. As the main result of this paper is the methodology, I believe the impact could be greatly improved with more details on the labelling strategy as well as evaluation of the features and thresholds used. Care needs to be made in the presentation of the results as the reader is required to make several assumptions to understand the figures. This makes it hard for me as a reviewer to evaluate the claims in the text as I cannot be sure I have interpreted the figures correctly. The figures in the supplementary are well described. I was able to download and successfully run CamoTSS with the default parameters, but ran into problems using the CTSS mode on the test data. I have detailed these comments below.

Response: thank you for the summary and appreciation of our work.

Major Comments:

1. Both PacBio and ONT are planning on releasing single cell versions of their technology. The authors should comment on the applicability of their tool on this coming technology or what/how much would need to be done to allow it to be used.

Response: Thanks for the suggestion. Here, we explored the potential applicability of the PacBio long-reads platform by downloading a PBMC dataset sequenced by SQ2 platform MAS-seq from PacBio (https://downloads.paccloud.com/public/dataset/MAS-Seq/DATA-SQ2-PBMC_5kcells/). After the genome alignment by pbmm2 (v1.12.0), we compared the distribution of position 0 of 5'scRNA-seq MAS-seq. One example was shown in the Fig. R6, which indicates the potential feasibility of using the MAS-seq data to detect TSS. However, template switch oligo (TSO) is a crucial technical element for CamoTSS to filter false positive TSSs, but reads in MAS-seq do not contain TSO, so it is hard to make CamoTSS applicable on the MAS-seq immediately. We plan to extend CamoTSS to support long reads in the next version.

Fig. R6 Two examples including CAMTA1 (Top) and VAMP3 (Bottom)

showing the distribution of position 0 of reads in 5' scRNA-seq (left) and MAS-seq (right).

2. Several of the figures and associated text leave a lot of room for assumption by the reader and it is likely I have misunderstood them, meaning I cannot fully verify the claims of the paper. Below is a list of areas where clarity can be improved which directly affect the claims. Smaller comments on the clarity of figures are under minor comments.

* The paper claims that the separation using TSS profiling is more consistent than the original paper however unless the reader finds the paper they can not see if this is more consistent or not. I suggest the authors include the original clustering so readers can make a direct comparison.

Response: Thank you for your comment. Yes, that was our original claim while as Reviewer 1 (point 1) suggested that such higher consistency is mainly observed in the coarse level annotation and this improvement will be less obvious with the finer resolution cell types. In this revision, we further clarified our finding that the primary cell variability represented differently at TSS level (S4 and S9 have largest difference) and gene level (R5 and R7 are most distinct subgroups). This suggests that the TSS level expression has complementary information to the gene level expression, presumably through gene regulatory patterns, which is well supported by our extended analyses showing the TF activities are highly concordant with TSS-based cell segregation (clusters S4 and S9). Details can be found in the description of Figure 3A-C and supplementary Figure S7-S9.

* Figure 2E supports the claim that samples across organs have high similarity. It would be good to see how the histogram looks without clustering when grouped by organs and cells as a comparison to see how much higher this clustering is.

Response: Thank you for your suggestion. According to your suggestion, we plot the heatmap without clustering (Fig. R7 and Fig. S10 in the revision) and found they are supportive to our original conclusion.

Fig. R7 Heatmaps based on Pearson's correlation coefficient of profile of all cell types and sorted by organs (left panel) and cell types (right panel).

* The paper claims that the expressed cell proportion of CTCF in NLH is significantly higher than NPC is significantly higher figure 3J, but no statistical test is shown.

Response: Thank you for the comment. As we focus on the variability across donors, we performed analysis on the donor level in Fig. 3J (Fig. 4J in the revision). This also means that we have a limited number of samples (n=3 for NLH and n=7 for NPC), resulting in low power to achieve statistical significance (e.g., with t-test and Wilcoxon rank-sum test). Instead, we showed this example as evidence for the same trend change (fold change = 1.235 between NLH vs NPC for proportion of cells expressing CTCF).

* It is stated that all significant TFs show a consistent trend and that they segregate the NLH and NPC samples. However Figure 3K doesn't show a striking difference in the clustering with only KLF6 looking like the only TF with a major difference. If this TF was removed from the analysis would the samples still cluster together. If you were to take the mean of each TF grouped by patient condition (removing patient _3 as an outlier) how many would have a significant difference between them?

Response: Thank you for the comment. First, we would like to admit that there

was an issue in our original manuscript where we did not consider the multiple binding frequency for each TF. Now we have corrected this and re-performed the Fisher exact test to examine if the binding frequency changes significantly between NLH/NPC groups for each TF; similarly, the significant TFs (FDR<0.01 for B cells and T cells) are shown in Fig. 3K. Second, in the original figure, indeed the difference was not shown clearly, partly because we used the expression level as the color scale, making the highly expressed KLF6 the most striking one. In the revision, we changed the presented value from mean expression to the expressed cell proportion and normalized it for each TF (row) into the range [0, 1]. Therefore, visually there are more TFs showing striking differences between NPC and NHL, e.g., GLIS3, THRA and THRB. Additionally, we used the fold change as a quantitative metric, indicated by asterisks in the updated Fig. 3K and R8A below (unfortunately, the sample size is too small to have enough statistical power as mentioned in the previous point). Specifically, we found 24 TFs showing >1.5 fold change or <0.6 fold change, 23 TFs with >1.2 & < 1.5 fold change or >0.6 & <0.8 fold change. The up- and down- regulated TFs were shown in red and blue, respectively. As suggested, we also examined the impact of removing KLF6 and patient_3 and found they do not affect the sample clustering (Fig. R8 B & C). Of note, for the hierarchical clustering, the two branches of any inner node can be ordered arbitrarily, so the three dendrograms are actually highly similar.

Fig. R8 Heatmap shows the hierarchical clustering of patients by the proportion of expressed cells of TFs that have significant differential binding frequency between NPC and NLH groups (n=10 patients). (A) Shown is all 10 patients and all significant TFs. (B) shown is 9 patients by removing patient_3 as potential outlier. (C) shown is all patients on all TFs but KLF6. The color in the heatmap means the proportion of expressed cells with rescaling to the range of 0 and 1 on row. The up- and down- regulated TFs were displayed in red and blue, respectively (NPC vs NLH). Blue ID **: fold change < 0.6; Blue ID *: 0.6 < fold change < 0.8; red ID *: 1.2 < fold change < 1.5; red ID **: fold change > 1.5.

3. The authors developed a combination of a linear regression model with 4 features with a CNN for the genomic sequences. As a key part of the paper, I believe further depth into the different choices made, and how they affect the performance of the model will greatly improve the paper. Below are my suggestions and queries regarding these models.

* ATAC peaks are theoretically not always specifically at the TSS region including the mentioned CTCF binding sites. This could potentially result in clusters not from real TSSs being incorrectly labelled as a positive. The authors should comment on this and explain how much the impact of this loose labelling may impact the performance of their model.

Response: Thank you for pointing out this potential issue. It's true that ATAC peaks appear in the location of chromatin accessibility, including but not limited to the region of transcription start site. But the other way around is generally more reliable that TSS regions have ATAC peaks and that is how we choose the positive and negative samples from the clusters of position 0 of reads, which makes the use of ATAC peaks more specific to TSS. We have added more explanations to the methods when defining positive and negative TSS samples (line 360 on p.13 and line 409 on p.15).

* I noticed that the number of positive and negative labels were close to being balanced (5560 and 5432 labels respectively). It is good that the authors report this information. I would like to know if it is a coincidence that the labels are balanced or was a method used to ensure a balanced training set? If so this should be detailed in the methods section of the paper.

Response: Thank you for the comment. It is not a coincidence. We subsample to keep a balanced training set. Corresponding description was added to the method section (p.14).

* It would be good to provide more details on the negative labelled clusters as they are currently an enigma but are an important part in training the model. How many of these were associated with a gene? For those that were annotated, where were they found along the gene body? Given that there are reads associated with negative clusters, this suggests transcriptional activity from where they are transcribed, why would they be associated with low ATAC peaks? As artifactual clusters are filtered out, what else could be causing the existence of these negative clusters (this would be good to explain in the introduction/discussion to detail the challenge of predicting TSS regions)?

Response: Thank you for your great suggestion. They are all associated with a gene (we fetch reads by using gene ID tagged in the bam file), so they are all located within the gene body. To clarify, in the original manuscript, we set the negative TSS to be associated with low ATAC peaks, while our intention was

defining them as not supported by ATAC peaks, so either intersecting with low confidence peaks or no peak at all (similar reason as our response to your first point in Comment #3). We have updated this definition explicitly while we found that there is only minimal increase from original analyses (59 in iPSC and 39 in DMFB that do not intersect with any ATAC peaks), so we keep our pre-trained model as it is. On the other hand, this expanded definition helps to increase 6,510 negative TSSs in the newly added PBMC data, largely because of the much lower coverage in the scATAC-seq data compared to bulk ATAC-seq.

For your last subquestion, our assumption is that the ATAC-seq data (especially bulk) gives reliable labels to the TSS candidates from 5' scRNA-seq data. Therefore, we anticipate that this filtering procedure in a supervised manner can remove most technical false positives. As reported, the technical features, e.g., un-encoded G and sequence features representing motifs are strong predictors for the filtering of these negative clusters.

* Figure 1E shows there is an enrichment of several other factors associated with TSS regions around their positive labels. This is evidence that many of their positive labels are correct. These graphs could also be broken down to show this distribution around TSSs detected in the CDS-exon and CDS-intron region (in response to comment a) and also negative labels (in response to comment c).

Response: Thank you for your great suggestion. As you suggest, we have broken down the TSS detected to CDS-exon, intron and negative labels to show the distribution of histone protein signals (Fig. R9 and Fig. 2E in the revision). These signals indicate the reliability of the TSSs detected by CamoTSS.

Fig. R9 The distributions of RNA POL2, H3K4me3, H3K27ac and H3K36me3 signals around the TSSs detected by us and the random regions produced by bedtools. RNA POL2, H3K4me3 and H3K27ac show enrichment around TSSs while H3K36me3 is enriched downstream of our TSSs.

*The CNN used random regions as negative labelled sequences. Therefore I would assume the model is learning the binding sites for particular transcription factors in the sample. If this assumption is correct it would mean that the model would not perform well in samples from other datasets reliant on other binding sites, or in other species. If this assumption holds, either this limitation needs to be discussed or to claim it is generalizable, an evaluation on a broad range of data needs to be shown.

Response: Thank you for your comments. Actually, here we used the same strategy to label the negative samples as the cluster model in the CNN model. Indeed, your suspicion is reasonable that the model may not generalizable to those binding motifs not seen in the training set. However, we expect that the patterns of binding sites from different samples (TFs) often have high similarity. On the one hand, most binding sites are similar for the TFs from the same family. On the other hand, Li et al⁴ have already reported the high conservation of binding sites among different species.

Here, to further demonstrate the robust generalization, we also performed out-of-distribution tests. Specifically, we took two TFs (i.e. CTCF; E2F6) as examples, where the sequences comprising binding sites of CTCF or E2F6 were

removed from the training dataset and then were used to predict sequences which only contain CTCF or E2F6. AUROC can reach 0.908 and 0.948 for CTCF and E2F6 knockout models, respectively (Fig. R10 and Fig. 2C in the revision), demonstrating the robustness of our sequence model.

Fig. R10 ROC curves for using samples which do not contain binding sites of the CTCF or E2F6 as training dataset to predict samples which only contain binding sites of the CTCF or E2F6.

* Clusters were filtered out if the inter-cluster distance was less than 300bp, in the methodology, clusters were merged if they were within 100bp distance from each other. In the event two clusters occur 200bp apart, are they both discarded, or is only one discarded? Was there any evaluation on the selection of these thresholds and does it greatly impact the performance of CamoTSS if they are altered and how? Similarly for the 15bp window size for CamoTSS?

Response: Thank you for the question and comments. When the two clusters occur 200bp apart, only the one with lower UMI count was discarded.

According to your suggestion, we evaluated the different inter-cluster distances from four aspects including the number of genes with alternative TSS, the number of TSS, the percentage of annotation TSS and distribution of scATAC-seq signal (Fig. R11 and Fig. S4 in the revision) and found that it did not greatly impact the quality of TSS detected by CamoTSS, though the number of TSSs and genes changes more linearly.

We also evaluated the window size for detecting CTSS in the CamoTSS by checking the number of CTSS, the percentage of annotated TSS (allowing +5/-5bp shift) and the CTSS intersection of different window size groups (Fig. R12 and Fig. S31 in the revision). The results show that the window size also won't largely impact the performance of CamoTSS.

Fig. R11 Evaluation of different inter-cluster distances. Line chart showing the number of genes with alternative TSS (A) and TSS (B) and percentage of annotated TSS (C) detected by CamoTSS in PBMC dataset. Peak plot displaying the scATAC-seq signal around TSS detected based on various inter-cluster distances (D).

A

window size	No. of CTSS	Percentage of annotated CTSS
15bp	2613	38.5%
30bp	2811	39.3%
60bp	2224	39.2%

Fig. R12 Evaluation for different window sizes including 15bp, 30bp and 60bp. (A). Statistic information for number of CTSS and percentage of annotated CTSS in different window sizes. (B). venn diagram showing the number of shared and unique CTSS among three window size groups.

* It would be good to include discussion or evaluation of the importance of the four features (five if including genomic sequence) used in the model. This can be done with many methods including but not limited to removing each feature and seeing how the performance of the model is affected.

Response: Thank you for your great suggestion. We used each feature or remaining features removing one feature from all features to evaluate the importance of five features (Fig. R13 and Fig. 2D in the revision). We found that the DNA sequence feature set (i.e. 32 weight features from CNN model; AUC=0.976) is the most predictive one, followed by the unencoded G percentage feature (AUC=0.974, Fig. R13). The coefficients of logistic regression display the consistent importance rank (Fig. R14 and Fig. S2 in the revision).

Fig. R13 All features (e.g. clusters features and sequence features), the combination dropping one feature in all features and each one feature were fed to the logistic regression model to do prediction. AUROC values are obtained

via 10-fold cross-validation.

Fig. R14 Bar plot showing feature importance ranking based on the coefficient in the logistic regression. All features were standardized at first.

*The inclusion of single-nucleotide resolution predictions in a TSS cluster is a great addition (CTSS). Can the authors comment on the likely accuracy of these predictions or include evaluation of how precise these are.

Response: Very good point; thanks. Unfortunately, there is no suitable evaluation criteria to assess the precision of CTSS. Here, we selected the percentage of annotated CTSS (i.e. comparing with reference annotation file), allowing +/- 5bp shift. As shown in the Fig. R15, the proportion of annotated CTSS is in a reasonable range (~39%).

window size	No. of CTSS	Percentage of annotated CTSS
15bp	2613	38.5%
30bp	2811	39.3%
60bp	2224	39.2%

Fig. R15 Evaluation for different window sizes including 15bp, 30bp and 60bp. Statistic information for number of CTSS and percentage of annotated CTSS in

different window sizes.

4. I was unable to run CTSS mode with the test data provided, getting the following error message “FileNotFoundError: [Errno 2] No such file or directory: 'CamoTSS_CTSS/count/afterfiltered.csv’”. If the test data is insufficient for this method it would be good for users to provide one.

Response: Thank you for helping testing our package. We have now fixed this issue and you are welcome to try it again. If you experience any other issues, please let us know in the GitHub issues or mention it in the future comments.

Minor Comments:

1. When clusters are annotated to genes, is there a maximum distance threshold used? If so, this should be listed in the methods.

Response: Actually not. At the beginning, we obtained reads for each gene by using the cigar tag “GX” for gene ID.

2. Below are comments related to the clarity of the figures and the associated text

* Figure 1B is hard to understand as many aspects of the figure are not labelled or described. Does C stand for cluster? Are the lines within the circles meant to represent a string of alignments and the red circles highlighting a read start? This should be added in the legend.

Response: Thank you for the suggestion. Yes, all of your descriptions are correct. We added them to the legend (p.3).

* In the legend for Figure 2A, it should be made clearer that the 3rd UMAP is the clusters from the RNA clustering.

Response: Thank you for the suggestion. We have now renamed the title of the 3rd UMAP in the Fig. 3A (revision version).

* Figure 2C mentions merged data. It should be made clearer if the merging was done in the original paper or in this paper.

Response: Thank you for the suggestion. The original paper contains 15 organs, so they performed two annotations: one for each organ, and another for the merging of 15 organs with higher resolution. So the merged data was done in the original paper. We have further clarified this in the text (Supplementary Fig. S7 in the revision).

* Figure 2E the heatmap dendrogram label columns are not labelled so it is not initially clear which column is cell types and which is organ.

Response: Thank you for the suggestion. We added the line to show which column is cell types and which is organ as shown in the Fig.3D and Supplementary Fig. S10.

* Figure 3D. TSS1 2 and 3 are not labelled on the figure, and the reader needs to look at a small transcript id to match them up. Additionally the vertical line hides the peaks making it harder to see differences.

Response: Thank you for the suggestion. We labelled the TSS1, 2 and 3 on Figure 4D and also removed the vertical line.

* Figure 3D. Is this a genome track plot of one cell or the coverage across all cells of each type? Make this clearer in the figure legend.

Response: Thank you for the suggestion. It is for all cells in a cell type; now the information has been added to the legend (Figure 4D in the revision).

* For readers that are less familiar with the biological background it would be good to include why CTCF is critical and well-studied and therefore chosen as an example.

Response: Thank you for the suggestion. The reason to select CTCF as an example was added to page 8.

* Figure 3E - Figure legend says this is for the LIST gene but the figure is labelled with LIMS. Make it clear if the violin plot contains the proportions of TSS1 from each cell.

Response: Thank you for pointing out - typo corrected now.

* There is no description for the colour legend in figure 3K

Response: Added; Thanks!

3. When running CamoTSS with the default parameters it produced an empty CTSS directory. If no CTSS output is expected then it would be less confusing for users to not have this directory created.

Response: Corrected; Thanks!

4. The output fourFeature.csv does not have clear column descriptions. Either delete this output if it is an intermediate file not meant to be seen or add column names so that advanced users could use this information.

Response: Column names added now; Thanks!

Reference

1. Aibar, S. *et al.* SCENIC: single-cell regulatory network inference and clustering. *Nat. Methods* **14**, 1083–1086 (2017).
2. Chen, Y.-P. *et al.* Single-cell transcriptomics reveals regulators underlying immune cell diversity and immune subtypes associated with prognosis in nasopharyngeal carcinoma. *Cell Res.* **30**, 1024–1042 (2020).
3. Moody, J. *et al.* Profiling of transcribed cis-regulatory elements in single cells. *bioRxiv* (2021) doi:10.1101/2021.04.04.438388.
4. Li, L., Zhang, S. & Li, L. M. Dual Eigen-modules of Cis-Element Regulation Profiles and Selection of Cognition-Language Eigen-direction along Evolution in Hominidae. *Mol. Biol. Evol.* **37**, 1679–1693 (2020).

Reviewer #1 (Remarks to the Author):

The authors have addressed my comments.

Reviewer #2 (Remarks to the Author):

The authors have addressed most of my concerns, but I would still like to see the application of CamoTSS to an independent cancer dataset. The current applications are a bit limited in scope. Showing convincing results for another cancer dataset will make the paper stronger and attract more potential users.

Reviewer #3 (Remarks to the Author):

In the response the authors have very comprehensively addressed all concerns I had as well as answering all questions that arose from reading their manuscript. They have updated their manuscript in response to the comments from me and the other reviewers, and I believe these changes make their already interesting manuscript even stronger. In addition they updated their software package, resolving the minor issues I had running the tool. I commend the authors on their responses, the comprehensiveness and effort with which they went to in their answers, and the transparency in how they changed their analysis from the original manuscript.

Reviewer 1: no more additional comments

Reviewer 2: page 1

Reviewer 3: no more additional comments

=====

Reviewer 1

The authors have addressed my comments.

Response: thank you for the affirmation of our reponse.

Reviewer 2

The authors have addressed most of my concerns, but I would still like to see the application of CamoTSS to an independent cancer dataset. The current applications are a bit limited in scope. Showing convincing results for another cancer dataset will make the paper stronger and attract more potential users.

Response: thank you for insisting on this suggestion. Indeed, it will further demonstrate the power of our method by adding one more independent cancer dataset. After a broader search of more cancer types, we found a recently published gastric cancer dataset¹. By focusing on the epithelial cells (n=8,485) which include 5,977 normal cells and 2,508 tumor cells, we performed CamoTSS for the alternative TSS analysis. Briefly, we found 453 genes with significant TSS shifting between normal and tumor cells, many of which play an essential role in cancer. In addition, the transcription factors regulating the TSS shift also have been reported to be involved in tumorigenesis. More details have been updated in *Alternative TSS usage in the tumor cell of gastric cancer* section (p.12).

Reviewer 3

In the response the authors have very comprehensively addressed all concerns I had as well as answering all questions that arose from reading their manuscript. They have updated their manuscript in response to the comments from me and the other reviewers, and I believe these changes make their already interesting manuscript even stronger. In addition they updated their software package, resolving the minor issues I had running the tool. I commend the authors on their responses, the comprehensiveness and effort with which they went to in their answers, and the transparency in how they changed their analysis from the original manuscript.

Response: thank you for the endorsement of our response.

Reference

1. Sun, K. et al. scRNA-seq of gastric tumor shows complex intercellular interaction with an alternative T cell exhaustion trajectory. *Nat. Commun.* 13, 4943 (2022).

Reviewer #2 (Remarks to the Author):

I would like to thank the authors for adding an additional cancer example. The paper is now much stronger with this additional demonstration of its utility. Congratulations!

Reviewer 1: no more additional comments

Reviewer 2: no more additional comments

Reviewer 3: no more additional comments

=====

Reviewer 2

I would like to thank the authors for adding an additional cancer example. The paper is now much stronger with this additional demonstration of its utility. Congratulations!

Response: thank you for the affirmation of our reponse.